# Does childhood experience of family victimization influence adulthood refusal of wife abuse? Evidence from rural Bangladesh

Rabiul Karim[1,2]☉*, Suchona Rahman[1‡], Hafijur Rahman[1‡], Tanzima Zohra Habib[1‡], Sadequl Arefin[1], Katarina Swahnberg[2]☉

1 Department of Social Work, University of Rajshahi, Rajshahi, Bangladesh, 2 Department of Health and Caring Sciences, Faculty of Health and Life Sciences, Linnaeus University, Kalmar, Sweden

☉ These authors contributed equally to this work.
‡ SR, HR and TZH also contributed equally to this work.
* Rabiul.Karim@lnu.se, rkarimsw@yahoo.com

**Data Availability Statement:** All relevant data are within the manuscript and its Supporting Information files.

## Abstract

This study examined how different forms of childhood family victimization are associated with the attitudinal (not actual action) refusal of wife abuse among women and men in rural Bangladesh. It included 1,929 randomly selected married women and men. Of the sample, 31.3% (*Men* = 49.3%, *Women* = 13.5%) attitudinally refused overall wife abuse, 38.5% (*Men* = 53.2%, *Women* = 23.8%) refused emotional abuse, 67.0% (*Men* = 82.5%, *Women* = 51.6%) refused physical abuse, 78.0% (*Men* = 88.6%, *Women* = 67.4%) refused abuse on wife's disobeying family obligations, and 32.3% (*Men* = 50.3%, *Women* = 14.6%) refused abuse on challenging male authority. Multivariate logistic regression revealed that the odds ratio (ORs) of the attitudinal refusal of overall wife abuse were 1.75 ($p$ = .041) for the childhood non-victims of emotional abuse and 2.31 ($p$ < .001) for the victims of mild emotional abuse, compared to the victims of severe emotional abuse. On the other hand, the ORs of the overall refusal of abuse were 1.84 ($p$ = .031) for the non-victims of physical abuse and 1.29 ($p$ = .465) for the victims of mild physical abuse, compared to the childhood victims of severe physical abuse. Data further revealed that the childhood non-victimization of physical abuse increased all types of attitudinal refusal of wife abuse, e.g., emotional abuse, physical abuse, abuse on disobeying family obligations, and abuse on challenging male authority. Compared to the childhood experiences of severe emotional abuse, data also indicated that childhood exposure to mild emotional abuse might increase the attitudinal refusal of wife abuse on a few issues, e.g., abuse on disobeying family obligations, abuse on challenging male authority, and physical abuse. It appeared that childhood experiences of family victimization greatly influence different types of attitudinal refusal of wife abuse. We argue that the issue of childhood victimization should be brought to the forefront in the discourse. We recommend that state machinery and social welfare agencies should expend significant efforts to stop child abuse within the family and in other areas of society in rural Bangladesh.

**Funding:** The authors acknowledge partial financial Supports from Rajshahi University, Swedish Research Council and Linnaeus University. Rabiul received partial funding from Rajshahi University. This fund was used for piloting the Study including instrument development and validation. Rabiul also received a fieldwork grant from Swedish Research Council as well as Katarina Received partial funding from Linnaeus University. These funds were used for this survey. There was No additional external funding received for this study. The funders had no role in study design, data collection and analysis, decision to publish, or preparation of the manuscript.

**Competing interests:** The authors have declared that no competing interests exist.

# Introduction

Attitudinal refusal of wife abuse among men and women is a crucial factor that practitioners attempt to enhance in the society for the primary prevention of intimate partner violence (IPV) in many low- and lower-middle-income countries. Wife abuse by the husband is the most common type of IPV perpetrated against women worldwide [1]. As the attitudinal refusal of wife abuse focuses on a mindset and not actual acts of abuse, it is a crucial aspect of IPV. Refusal/acceptance of wife abuse is also a widely talked about issue in the discourse of the primary prevention of IPV against women. Both scholars and practitioners advocate for community-based interventions for enhancing refusal of wife abuse among the citizens [2–4]. Previous studies show that people's attitude related to wife abuse is a significant feature of IPV perpetrated against women [5–9]. This is because, in patriarchal societies, people believe that a husband has a right to 'punish' his wife under some conditions, e.g., if a wife argues with her husband [10–12]. This attitude is widely prevalent in Bangladesh [12]. Husbands not only abuse their wives, it is also endorsed by the wife, family members, and in-laws. This attitude indicates why a society rationalizes 'the privilege of a husband' to abuse his wife [10, 12].

Previous studies conducted in the African, South Asian, and Middle-eastern countries demonstrate that people's attitude toward wife abuse is an important feature of IPV against women [5–9, 11, 13, 14]. However, there is a lack of studies focusing particularly on the refusal of wife abuse. In order to prevent wife abuse in low- and lower-middle-income countries, we also need to study the people who reject wife abuse. Hence, it is essential to explore the factors that contribute to the refusal of wife abuse in different cultural contexts.

The current study is an attempt to explore the phenomenon in rural Bangladesh. The country is situated in South Asia, comprising 162 million people, where nearly 21.8% live below the poverty line–earning less than two USD a day [15]. The mainstream population of the country is traditionally patriarchal. Gender inequality is very high. The country ranks 129 out of 162 countries in the Gender Inequality Index (GII) [16]. The general aspects of gender inequality include women's lack of access to economic resources, employment, and higher education; male control over women's life choices; male-dominated social institutions; and widespread violence against women [10, 17]. In rural Bangladesh, wife abuse is deeply rooted in the patriarchal family institutions [10, 18]. In the rural areas, women are perceived as the dependent members of the households, whereas men are expected to be the economic providers. Therefore, men own most of the properties and become the head of the household. On the other hand, married women are expected to perform home-making chores, obey the husband, and take care of family members [19]. These patriarchal norms/attitudes provide context-specific reasons to rationalize 'the privilege of a husband' to control and abuse his wife. Therefore, the attitudinal refusal of wife abuse is quite low in the country.

An earlier study indicated that only 20% of the married women in rural Bangladesh attitudinally refused the notion that a husband has a right to beat his wife [10]. This is quite relevant to understand the extent of wife abuse in the country, where 80% of married women experience physical or sexual abuse from their husbands [17]. Both abusers and victims view husbands' violent behaviors against their wives as a 'common act' [12]. Violent husbands often try to explain their actions by pointing out their 'wives' faults' [12, 20]. Many wives also perceive these abusive acts as 'normal.' Therefore, we need to enhance the attitudes toward refusal of wife abuse among both women and men. In order to prevent the problem effectively, it is imperative to explore the factors influencing such attitudes against wife abuse in different social contexts. However, to our knowledge, no previous studies focus on the factors associated with the refusal of wife abuse in Bangladesh. The current study examines how childhood experience of family victimization is related to different forms of adulthood attitudes toward refusal of wife abuse in rural Bangladesh.

## Theoretical perspective

To understand the childhood learning of abusive behaviors and their subsequent implications for beliefs and attitudes toward wife abuse in adulthood, this study adopted the conceptual underpinnings of social learning theory [21, 22]. This theory explains how children acquire information on the use of violence through observational learning [21], where the family-of-origin is regarded as the 'training ground' for learning abusive behaviors [23]. Since it is argued that wife abuse is a learned behavior [24, 25], it can also be prevented successfully through appropriate educational and social control mechanisms [3]. To do so, however, we need to understand the issue within the contexts of a specific social system.

Children who experience familial abuses or who witness inter-parental abusive events are more likely to engage in abusive acts when they grow up [25]. According to Gelles [25], through their abusive and submissive behaviors, parents teach children to engage in such behavior. Children also learn the appropriateness of certain abusive acts from their parents [24, 25]. In most cases, the family is the relationship where an individual experiences the first abuse in his/her life. A family not only teaches one to engage in violent behavior, but it also provides a justification for the use of such abusive behaviors [25].

Studies have consistently shown that childhood exposure to parental abuse is linked to their adulthood beliefs/behaviors related to wife abuse [23, 24], although little is known about why many adults who were exposed to abusive childhood events do not continue the same pattern with their partners [26]. It is possible that higher education and an awareness of the risks associated with abuse, a person's self-esteem, relationship quality, meaningful victim support system, gender-equitable environment, and suitable social control mechanism may de-motivate people to continue the abuse in adulthood even if they have experienced and internalized the abuse in their childhood [27–29]. A few studies based in Bangladesh have focused on the social learning of wife abuse and its social acceptance among people [7, 30]. However, the association between childhood non-experiences of family victimization with different forms of attitudinal refusal of wife abuse is relatively overlooked in previous studies conducted in Bangladesh. Hence, this study aims to enhance our understanding of what influence childhood learning of abuses has on the refusal of wife abuse in adulthood.

Previous studies mainly considered the attitude toward wife abuse as a mono-dimensional category [7, 30]; however, a recent study indicates that there may be at least four dimensions where a person may justify/refuse wife abuse [31]. The four dimensions are as follows: acceptance/refusal of emotional abuse, acceptance/refusal of physical abuse, acceptance/refusal of abuse on the grounds of wife's disobeying family obligations, and acceptance/refusal of abuse on wife's challenging male family authority [31]. Therefore, it is crucial to understand how childhood experiences of family victimization influence various types of attitudinal refusal of wife abuse in rural Bangladesh. Preceding studies were also limited in that they conceptualize childhood victimization as a single category [7, 32–35], although it may also have different dimensions, e.g., childhood exposure to emotional and physical abuses with varying degrees of intensity, e.g., non-exposed, mildly exposed, and severely exposed [36]. Therefore, it is crucial to examine how different forms of childhood family victimization are associated with different types of attitudinal refusal of wife abuse.

## Objectives of the study

The aim of this study is to examine how different types of childhood experiences of family victimization (or non-victimization) are associated with the attitudinal refusal of wife abuse among both women and men in rural Bangladesh. Specifically, the study will examine the associations between childhood family experiences of emotional and physical (non-) victimizations and the adulthood attitudinal refusal of wife abuse in its various domains.

## Methods and materials

### Study design

The current study used baseline data of a cluster-randomized control trial (C-RCT) 'Community-based prevention of domestic violence among *Bengali*, *Garo*, and *Santal* ethnic communities in rural Bangladesh: A cross-cultural study.' The survey was conducted from February to May 2019. The main study investigates the efficiency of the intervention. Attitude toward wife abuse was used as one of the outcome measures of the study. Although the main study is C-RCT in nature, the current data represent a retrospective cohort design [37]. Both the exposure and response data were collected from the same respondents during adulthood, while they retrospectively reported their childhood experiences of abuse.

### Sites under study

The study was conducted in 24 purposively selected villages located in different parts of Bangladesh. We included mainstream *Bengali* as well as ethnic minority *Santal* and *Garo* villages, respectively. The mainstream *Bengali* villages were located in two sub-districts in northwest Bangladesh. The ethnic *Santal* villages were selected from another northwest sub-district, where the indigenous people have settled for about 300 years. On the other hand, the ethnic *Garo* villages were selected from a northeast sub-district where they lived for more than 500 years. The mainstream *Bengali* and the ethnic minority *Santal* villages are patriarchal, whereas the ethnic *Garo* society follows matrilineal (matriarchal) traditions. In the *Garo* villages, women are traditionally treated as the heads of the households; and all of the family properties are also passed down to the female line. On the other hand, men own most of the household properties and rule the families in both *Bengali* and *Santal* villages. Due to their veiled seclusion, women's physical mobility is quite low in the *Bengali* villages. However, women enjoy greater freedom to move in public spheres in both *Garo* and *Santal* villages. There are about 68,000 villages in Bangladesh; therefore, it was beyond our time and budget to embrace a statistically representative number of villages. However, we selected our study participants from the villages using a random sampling method.

### Study participants

The sample included currently married men and women aged 16−60-years-old. Due to our lack of preparation and appropriate facilities, we excluded individuals with emotional disorders and physical disabilities (e.g., deaf/mute). By taking into account the findings from our pilot study conducted in selected villages, we computed the sample size for the baseline survey. Considering the prevalence of sexual abuse in a village ($p = 0.167$), the minimum sample size was estimated at 1,854 using a formula ($n = z^2_{\alpha/2}\,p(1\text{-}p)/E^2$, where p = Proportion; $\alpha = 0.05$; therefore, $z_{\alpha/2} = 1.96$; $E = p/10$) [38]. To account for possible non-consent and drop out, we created a sample pool, including 10% over samples. In total, we approached 1,968 persons. We used a cluster sampling procedure in order to select our study participants.

   At first, we identified the study villages. After selecting a village, we collected an updated list of households. Each of the villages consists of roughly 100−300 households. We randomly identified the first household in a selected village. In order to form a cluster, we then included 81 additional households close to the first household. These households were the most physically nearest households from the household we selected at first. To do so, we conducted a quick household mapping in the area. After that, we randomly assigned half of the households for male respondents and the other half for female respondents. As a final point, we approached one respondent from a household for face-to-face interviews. If a household had

more than one eligible respondent, we randomly selected one. In total, 1,929 respondents (961 men and 968 women) completed the questionnaire. The response rate was 98.02%. Reasons for non-responses were mainly related to the participants' lack of time.

## Response variable

**Attitudinal refusal of wife abuse.** The refusal of wife abuse by women and men was measured using a 6-item scale validated in rural Bangladesh [31]. We assessed the number of abusive events refused by a respondent. The scale included six questions on the contextual reasons where people may refuse/justify a husband's right to abuse his wife in Bangladesh (see S1 Table). We asked the respondents their opinions on whether a husband can abuse his wife under given conditions, for example, 'if a wife fails to prepare meals on time.' There were four response categories: 0 = no (refused abuse), 1 = yes–emotional abuse, 2 = yes–physical abuse, and 3 = yes–both emotional and physical abuses. We provided examples of both emotional abuse (e.g., humiliation, insults, verbal rebuke, cursing, displaying anger, threatening to beat, etc.) and physical abuse (e.g., slapping, grabbing mouth, punching, kicking, beating with a fist/stick, burning, etc.). The scale produced two variables: refusal of emotional abuse and refusal of physical abuse. In order to compute the variables (refusal of emotional abuse and refusal of physical abuse), we recoded the responses. We derived the refusal of emotional abuse variable after merging 'yes–emotional abuse' and 'yes–both emotional and physical abuse' responses. Then again, we got the refusal of physical abuse by adding 'yes–physical abuse' and 'yes–both emotional and physical abuses' responses. In this study, the scale appeared to be internally consistent and highly reliable ($\alpha$ = 0.87).

The 6-item scale also had two underlying factors: disobeying family obligations and challenging male authority. The first factor, 'disobeying family obligations,' measured the refusal of abuse in the scenarios where the wife disobeys her family obligations (e.g., failing to prepare tasty meals). In contrast, the second factor, 'challenging male authority,' assessed the refusal of abuse where the wife confronts male authority (e.g., arguing with him) [31].

Therefore, we constructed five binary variables: (a) refusal of overall wife abuse–where the respondent refused both emotional and physical abuse for all six questions, (b) refusal of emotional abuse, (c) refusal of physical abuse, (d) refusal of abuse on disobeying family obligations, and (e) refusal of abuse on challenging male authority. We considered a refusal of abuse when the respondent answered with a 'no' to all the responses for the relevant items.' After that, we scored the variables: 1 = refused abuse, and 0 = did not refuse.

## Exposure variable

**Childhood family victimization.** We used two variables: (a) childhood experiences of emotional abuse and (b) childhood experiences of physical abuse. Both the variables had three categories: 0 = none, 1 = mild abuse, and 2 = severe abuse. We only assessed those who experienced abuse, not those who witnessed inter-parental abuse. A revised version of the NorVold Abuse Questionnaire was used to assess childhood exposure to victimization [36]. We included six sets of questions on the experiences of emotional and physical abuses (see S1 Table). We considered various emotional and physical abuses experienced by the study participants in their families during childhood. The notable feature of the NorVold Abuse scale is that it specifies the severity of abuse, namely mild abuse and severe abuse [36].

Regarding the nature of abuse, we considered some abusive items as 'mild abuse' and others as 'severe abuse.' For example, questions on mild emotional abuse included: 'has anybody in your family displayed anger or hatred toward you?' and for severe emotional abuse, questions were posed on threatening behaviors such as 'has anyone threatened to kill or injure you seriously.' On the other hand, questions on mild physical abuse included: 'has anyone twisted

your arm/hair or slapped you,' while severe physical abuse was questioned with: 'has anyone kicked or beaten you with a stick or something else.' Each of the questions was rated with the responses: '0 = no,' '1 = yes.' Finally, we constructed both the childhood abuse variables by considering relevant severity scores, e.g., 0 = none (non-exposed), 1 = mild (exposed to mild abuse only), and 2 = severe (exposed to any severe abuse–this also included those who experienced both severe and mild cases of abuse). In this study, the NorVold Abuse Questionnaire demonstrated a very high-reliability score (α = .86).

## Controlled variables

We also included controlled variables in the analysis. We controlled for a community level variable (e.g., ethnicity), a family level variable (e.g., family structure), and a few individual-level variables such as respondents' gender, age, education, and income.

The participants were divided into three categories based on ethnicity: *Garo*, *Santal*, and *Bengali*. The *Bengalis* are the mainstream ethnic community in the country. They are traditionally patriarchal. In contrast, the *Garos* are an ethnic minority community upholding matrilineal traditions, whereas the *Santals* are a patriarchal indigenous community living in Bangladesh. Gender had two socially constructed categories: men and women. Age was classified into four levels: 16–25 years, 26–35 years, 36–45 years, and 46–60 years.

Education was measured using the actual number of years of schooling. According to the typical categorization of educational attainment in Bangladesh, the score was later transformed into four levels: No schooling, Primary (1–5 years of schooling), Secondary (6–10 years of schooling), and Higher (Secondary School Certificate examination or above).

Monthly income was measured on a 3-point response category, ranging from 1 = no income (earning no cash income/salary), 2 = earning less than 7,000 Taka per month, and 3 = earning 7,000 Taka or above per month. Considering a poverty line income categorization procedure [39] and people's current socio-economic conditions in rural Bangladesh, the personal monthly income was classified into these three above-mentioned contextual levels.

Family structure had two categories: Nuclear (having husband, wife, and/or unmarried children); and Extended (including husband, wife, and/or married adult children/in-laws).

## Data collection

We used a structured questionnaire for face-to-face interviews. This allowed the interviewer to interact with the respondents and resulted in better quality responses. The study participants were personally contacted. Four graduates in social work (two males and two females) were employed to collect the data. Male interviewers questioned the male respondents, while female interviewers questioned the female respondents. The interviewers underwent training on ethical, safety, and technical issues related to data collection. They also received training on how to provide support to any abused respondents who sought help. We emphasized the establishment of rapport with the respondents. This was done by asking less sensitive questions first, which allowed the respondents to adapt more easily to the sensitive issues. Before having an interview, we explained the study protocols to each participant.

## Analytical strategies

The purpose of the data analysis was to examine the association between childhood exposure to family victimization and adulthood refusal of wife abuse. Descriptive statistics of the variables were produced, which provided a profile of the study participants. The response variable (refusal of wife abuse) was measured as binary categories; therefore, we employed logistic regression to assess the association between the response and exposure variables [40, 41].

Coefficients of logistic regression with log link function, exp*(β)*, produced Odds Ratio (OR), which facilitated the explanation of regression results. We used ORs to report associations between exposures and outcomes [40]. At first, bivariate logistic regression was employed in order to identify the significant independent variables. Subsequently, multivariate logistic regression was performed to test the factors influencing the response variable. The data analysis was conducted using SPSS 23.0 software [42].

### Ethical procedures

The study was conducted according to the operational guidelines and procedures recommended by the World Health Organization for conducting research on violence against women [43]. Ethical guidelines for public health research were also considered [44]. The study protocol was approved by the Ethics Review Committee at the Faculty of Social Sciences, University of Rajshahi, Bangladesh. We obtained informed consent from the study participants. Upon consent, we asked the study participants to suggest a suitable place and time so that the data collection process could take place in private. The participants were informed that they might find some questions uncomfortable. They were also reminded that their participation was entirely voluntary, and they had no obligation to complete the interview and could drop out of the interview at any time without any further explanation. Anonymity and confidentiality of the interviews were maintained. We also informed the participants about our intervention. We introduced the nearest domestic violence support services to participants, who disclosed experiences of abuse and sought support.

## Results

### Sample characteristics

We presented the socio-demographic characteristics of the study participants in Table 1. Nearly half (50.2%) of them were women. The majority (38.7%) of the participants belonged to a younger (26–35 years) age group. Of the sample, 33.2% were from *Garo*, 33.2% from *Santal*, and 33.6% from *Bengali* ethnic communities. By education, 42.5% had attained primary education, 29.8% had secondary education, and only 20.0% had a higher level of education. A majority of them (41.3%) were day laborers; and 37.5% earned a monthly income of BDT 7000/above (equivalent to US$30). Our data further show that there were gender differences in the socio-economic status of the respondents (see Table 1).

### Childhood experience of family victimization

The respondents widely experienced childhood family victimization, where men appeared to be less exposed to such abuses. Table 1 reveals that 14.8% of the participants (*Men* = 2.5%, *Women* = 26.9%) were exposed to severe emotional abuse, 67.5% (*Men* = 66.4%, *Women* = 26.9%) were exposed to mild emotional abuse, and only 17.7% (*Men* = 31.3%, *Women* = 4.3%) were not exposed to any emotional abuse, respectively. Our data also show that 23.9% (*Men* = 18.9%, *Women* = 28.9%) of the study participants were exposed to severe physical abuse during childhood, 42.9% of them (*Men* = 37.3%, *Women* = 48.4%) were exposed to mild physical abuse, and only 33.2% (*Men* = 43.9%, *Women* = 22.7%) were not exposed to any physical abuse within their family of origin, respectively (see Table 1).

### Prevalence of the refusal of wife abuse

Data revealed that the prevalence of attitudinal refusal of wife abuse was quite low among the sample. The rates of refusing wife abuse appeared to be different regarding the different

**Table 1. Prevalence of the different types of attitudinal refusal of wife abuse by gender.**

| | Sample Profile | | |
| --- | --- | --- | --- |
| | Total | Men | Women |
| | N = 1929 (%) | N = 960 (%) | N = 969 (%) |
| **Gender** | | | |
| Men | 960 (49.8) | - | - |
| Women | 969 (50.2) | - | - |
| **Ethnicity** | | | |
| Garo | 640 (33.2) | 318 (33.1) | 322 (33.2) |
| Santal | 640 (33.2) | 319 (33.2) | 321 (33.1) |
| Bengali | 649 (33.6) | 323 (33.6) | 326 (33.6) |
| **Age** | | | |
| 16–25 | 301 (15.6) | 34 (3.5) | 267 (27.6) |
| 26–35 | 746 (38.7) | 300 (31.3) | 446 (46.0) |
| 36–45 | 711 (36.9) | 487 (50.7) | 224 (23.1) |
| 46–60 | 171 (08.9) | 139 (14.5) | 32 (3.3) |
| **Schooling** | | | |
| None | 149 (07.7) | 51 (5.3) | 98 (10.1) |
| Primary | 820 (42.5) | 457 (47.6) | 363 (37.5) |
| Secondary | 575 (29.8) | 228 (30.0) | 287 (29.6) |
| Higher | 385 (20.0) | 164 (17.1) | 221 (22.8) |
| **Monthly income** | | | |
| No income | 423 (21.9) | 2 (0.2) | 421 (43.4) |
| Below BDT 7000 | 783 (40.6) | 177 (18.4) | 398 (41.1) |
| BDT 7000/above | 723 (37.5) | 781 (81.4) | 150 (15.5) |
| **Family type** | | | |
| Nuclear | 1411 (73.1) | 738 (76.9) | 673 (69.5) |
| Extended | 518 (26.9) | 222 (23.1) | 296 (30.5) |
| **Childhood exposure to emotional abuse** | | | |
| None | 341 (17.7) | 299 (31.1) | 42 (4.3) |
| Mild | 1303 (67.5) | 637 (66.4) | 666 (68.7) |
| Severe | 285 (14.8) | 24 (2.5) | 261 (26.9) |
| **Childhood exposure to physical abuse** | | | |
| None | 641 (33.2) | 421 (43.9) | 220 (22.7) |
| Mild | 827 (42.9) | 358 (37.3) | 469 (48.4) |
| Severe | 461 (23.9) | 181 (18.9) | 280 (28.9) |
| **Attitudinal refusal of wife abuse** | | | |
| Refused overall wife abuse (yes)[1] | 604 (31.3) | 473 (49.3) | 131 (13.5) |
| Refused emotional abuse (yes)[2] | 742 (38.5) | 511 (53.2) | 231 (23.8) |
| Refused physical abuse (yes)[3] | 1292 (67.0) | 792 (82.5) | 500 (51.6) |
| Refused abuse on disobeying obligations (yes)[4] | 1504 (78.0) | 851 (88.6) | 653 (67.4) |
| Refused abuse on challenging male authority (yes)[5] | 624 (32.3) | 483 (50.3) | 141 (14.6) |

[1]Refusal of both emotional and physical abuses on all of the six items included in the scale

[2]Refusal of emotional abuse on all of the six items included in the scale

[3]Refusal of physical abuse on all of the six items included in the scale

[4]Refusal of all of the abuses on the three relevant items representing disobeying family obligations

[5]Refusal of all of the abuses on the three relevant items representing challenging male authority (see S1 Table)

domains. The attitudinal refusal of different forms of wife abuse also appeared to be less among women than men. Table 1 shows that only 31.3% (*Men* = 49.3%, *Women* = 13.5%) of the respondents refused overall wife abuse, 38.5% (*Men* = 53.2%, *Women* = 23.8%) refused emotional abuse, 67.0% (*Men* = 82.5%, *Women* = 51.6%) refused physical abuse, 78.0% (*Men* = 88.6%, *Women* = 67.4%) refused abuse on wife's disobeying family obligations, and 32.3% (*Men* = 50.3%, *Women* = 14.6%) refused abuse on wife's challenging male family authority.

## Factors influencing the refusal of wife abuse

Table 2 presents the results of bivariate cross-tabulations between exposure/control variables and the response variables. It revealed that childhood non-victims were more likely to refuse different forms of wife abuse. The rates for refusal of overall wife abuse, refusal of emotional abuse, refusal of physical abuse, refusal of abuse on disobeying family obligations, and refusal on challenging male authority were appeared to be higher among the non-victims and mild-victims than the victims of severe emotional/physical abuses. Data also indicate that the refusal of wife abuse was higher among older respondents than the younger respondents. There were also ethnic, educational, and income differences in the refusal of wife abuse.

Multivariate binary logistic regressions were performed to predict how the exposure variables contributed to the different types of attitudinal refusal of wife abuse in the sample. However, to identify the significant exposure/control variables influencing the response variable, we first conducted bivariate binary logistic regression. This helps us to select the variables to be considered for further multivariate analysis. Table 3 represents the crude (unadjusted) odds ratio (ORs) for different forms of the refusal of wife abuse. It shows that, except for family structure, other variables were significantly associated with the response variables. Therefore, we included these significant variables in the multivariate models.

The results of multivariate logistic regression analysis are presented in Table 4. It represents the adjusted ORs for five different models explaining: (a) refusal of overall wife abuse, (b) refusal of emotional abuse, (c) refusal of physical abuse, (d) refusal of abuse on disobeying family obligations, and (e) refusal of abuse on challenging male authority.

**Refusal of overall wife abuse.** Data revealed that the odds ratio (ORs) for the refusal of overall wife abuse was 1.75 ($p$ = .041) for the childhood non-victims of emotional abuse and 2.31 ($p$ < .001) for the victims of mild emotional abuse compared to the victims of severe emotional abuse. It also shows that the ORs for the refusal of overall abuse were 1.84 ($p$ = .031) for the childhood non-victims of physical abuse and 1.29 ($p$ = .465) for the childhood victims of mild physical abuse, compared to the childhood victims of severe physical abuse within the family (see Table 4).

**Refusal of emotional abuse.** Table 4 reveals that the ORs for the refusal of emotional abuse were 0.81 ($p$ = .323) for the childhood non-victims of emotional abuse and 0.95 ($p$ = .773) for the victims of mild abuse, compared to the victims of severe abuse. It also shows that the ORs for the refusal of emotional abuse were 1.66 ($p$ < .001) for the childhood non-victims of physical abuse and 1.34 ($p$ = .028) for the victims of mild abuse, compared to the victims of severe physical abuse.

**Refusal of physical abuse.** Data show that the ORs for the refusal of physical wife abuse was 0.93 ($p$ = .742) for the childhood non-victims of emotional abuse and 1.78 ($p$ < .001) for the victims of mild emotional abuse compared to the victims of severe emotional abuse. It further reveals that the ORs for the refusal of physical wife abuse was 2.05 ($p$ < .001) for the childhood non-victims of physical abuse and 1.62 ($p$ < .001) for the victims of mild physical abuse, compared to the childhood victims of severe physical abuse within their family of origin (see Table 4).

**Table 2. Socio-demographic profile and bivariate differences in the different forms of attitudinal refusal of wife abuse, N = 1929.**

| | Refused overall (all types of) wife abuse[1] | | | Refused emotional abuse[1] | | | Refused physical abuse[1] | | | Refused wife abuse on disobeying obligations[1] | | | Refused wife abuse on challenging male authority[1] | | |
|---|---|---|---|---|---|---|---|---|---|---|---|---|---|---|---|
| | Total N = 604 (%) | Men N = 473 (%) | Women N = 131 (%) | Total N = 742 (%) | Men N = 511 (%) | Women N = 231 (%) | Total N = 1292 (%) | Men N = 792 (%) | Women N = 500 (%) | Total N = 1504 (%) | Men N = 851 (%) | Women N = 653 (%) | Total N = 624 (%) | Men N = 483 (%) | Women N = 141 (%) |
| **Childhood emotional abuse** | | | | | | | | | | | | | | | |
| None | 152 (44.6) | 145 (48.5) | 7 (16.7) | 170 (49.9) | 159 (53.2) | 11 (26.2) | 249 (73.0) | 231 (77.3) | 18 (42.9) | 285 (83.6) | 258 (86.3) | 27 (64.3) | 157 (46.0) | 150 (50.2) | 7 (16.7) |
| Mild | 428 (32.8) | 322 (50.5) | 106 (15.9) | 498 (38.2) | 345 (54.2) | 153 (23.0) | 925 (71.0) | 547 (85.9) | 378 (56.8) | 1038 (79.7) | 573 (90.0) | 465 (69.8) | 440 (33.8) | 327 (51.3) | 113 (17.0) |
| Severe | 24 (8.4) | 6 (25.0) | 18 (6.9) | 74 (26.0) | 7 (29.2) | 67 (25.7) | 118 (41.4) | 14 (58.3) | 104 (39.8) | 181 (63.5) | 20 (83.3) | 161 (61.7) | 27 (9.5) | 6 (25.0) | 21 (8.0) |
| **Childhood physical abuse** | | | | | | | | | | | | | | | |
| None | 275 (42.9) | 244 (58.0) | 31 (14.1) | 302 (47.1) | 257 (61.0) | 45 (20.5) | 491 (76.6) | 376 (89.3) | 115 (52.3) | 536 (83.6) | 394 (93.6) | 142 (64.5) | 282 (44.0) | 249 (59.1) | 33 (15.0) |
| Mild | 228 (27.6) | 159 (44.4) | 69 (14.7) | 302 (36.5) | 172 (48.0) | 130 (27.7) | 549 (66.4) | 303 (84.6) | 246 (52.5) | 657 (79.4) | 317 (88.5) | 340 (72.5) | 236 (28.5) | 162 (45.3) | 74 (15.8) |
| Severe | 101 (21.9) | 70 (38.7) | 31 (11.1) | 138 (29.9) | 82 (45.3) | 56 (20.0) | 252 (54.7) | 113 (62.4) | 139 (49.6) | 311 (67.5) | 140 (77.3) | 171 (61.1) | 106 (23.0) | 72 (39.8) | 34 (12.1) |
| **Ethnicity** | | | | | | | | | | | | | | | |
| Garo | 233 (36.4) | 179 (56.3) | 54 (16.8) | 265 (41.4) | 188 (59.1) | 77 (23.9) | 470 (73.4) | 272 (85.5) | 198 (61.5) | 504 (78.8) | 285 (89.6) | 219 (68.0) | 240 (37.5) | 181 (56.9) | 59 (18.3) |
| Santal | 221 (34.5) | 172 (53.9) | 49 (15.3) | 253 (39.5) | 183 (57.4) | 70 (21.8) | 436 (68.1) | 266 (83.4) | 170 (53.0) | 471 (73.6) | 282 (88.4) | 189 (58.9) | 232 (36.3) | 179 (56.1) | 53 (16.5) |
| Bengali | 150 (23.1) | 122 (37.8) | 28 (8.6) | 224 (34.5) | 140 (43.3) | 84 (25.8) | 386 (59.5) | 254 (78.6) | 132 (40.5) | 529 (81.5) | 284 (87.9) | 245 (75.2) | 152 (23.4) | 123 (38.1) | 29 (8.9) |
| **Age** | | | | | | | | | | | | | | | |
| 16–25 | 56 (18.6) | 12 (35.3) | 44 (16.5) | 86 (28.6) | 16 (47.1) | 70 (26.2) | 170 (56.5) | 22 (64.7) | 148 (55.4) | 233 (77.4) | 32 (94.1) | 201 (75.3) | 58 (19.3) | 12 (35.3) | 46 (17.2) |
| 26–35 | 223 (29.9) | 160 (53.3) | 63 (14.1) | 275 (36.9) | 167 (55.7) | 108 (24.2) | 501 (67.2) | 258 (86.0) | 243 (54.5) | 570 (76.4) | 267 (89.0) | 303 (67.9) | 229 (30.7) | 162 (54.0) | 67 (15.0) |
| 36–45 | 255 (35.9) | 236 (48.5) | 19 (8.5) | 301 (42.3) | 255 (52.4) | 46 (20.5) | 503 (70.7) | 409 (84.0) | 94 (42.0) | 561 (78.9) | 432 (88.7) | 129 (57.6) | 262 (36.8) | 241 (49.5) | 21 (9.4) |
| 46–60 | 70 (40.9) | 65 (46.8) | 5 (15.6) | 80 (46.8) | 73 (52.5) | 7 (21.9) | 118 (69.0) | 103 (74.1) | 15 (46.9) | 140 (81.9) | 120 (86.3) | 20 (62.5) | 75 (43.9) | 68 (48.9) | 7 (21.9) |
| **Schooling** | | | | | | | | | | | | | | | |
| None | 33 (22.1) | 25 (49.0) | 8 (8.2) | 60 (40.3) | 33 (64.7) | 27 (27.6) | 75 (50.3) | 37 (72.5) | 38 (38.8) | 106 (71.1) | 46 (90.2) | 60 (61.2) | 34 (22.8) | 25 (49.0) | 9 (9.2) |
| Primary | 265 (32.3) | 233 (51.0) | 32 (8.8) | 310 (37.8) | 250 (54.7) | 60 (16.5) | 555 (67.7) | 379 (82.9) | 176 (48.5) | 616 (75.1) | 401 (87.7) | 215 (59.2) | 276 (33.7) | 242 (53.0) | 34 (9.4) |
| Secondary | 163 (28.3) | 133 (46.2) | 30 (10.5) | 207 (36.0) | 138 (47.9) | 69 (24.0) | 384 (66.8) | 246 (85.4) | 138 (48.1) | 454 (79.0) | 257 (89.2) | 197 (68.6) | 168 (29.2) | 134 (46.5) | 34 (11.8) |
| Higher | 143 (37.1) | 82 (50.0) | 61 (27.6) | 165 (42.9) | 90 (54.9) | 75 (33.9) | 278 (72.2) | 130 (79.3) | 148 (67.0) | 328 (85.2) | 147 (89.6) | 181 (81.9) | 146 (37.9) | 82 (50.0) | 64 (29.0) |
| **Monthly income** | | | | | | | | | | | | | | | |
| No income | 68 (16.1) | 2 (100.0) | 66 (15.7) | 127 (30.0) | 2 (100.0) | 125 (29.7) | 215 (50.8) | 2 (100.0) | 213 (50.6) | 310 (73.3) | 2 (100.0) | 308 (73.2) | 71 (16.8) | 2 (100.0) | 69 (16.4) |
| Below BDT 7000 | 227 (29.0) | 182 (52.4) | 45 (10.3) | 278 (35.5) | 199 (57.3) | 79 (18.1) | 491 (62.7) | 271 (78.1) | 220 (50.5) | 564 (72.0) | 295 (85.0) | 269 (61.7) | 240 (30.7) | 190 (54.8) | 50 (11.5) |
| BDT 7000/above | 309 (42.7) | 289 (47.3) | 20 (17.9) | 337 (46.6) | 310 (50.7) | 27 (24.1) | 586 (81.1) | 519 (84.9) | 67 (59.8) | 630 (87.1) | 554 (90.7) | 76 (67.9) | 313 (43.3) | 291 (47.6) | 22 (19.6) |
| **Family type** | | | | | | | | | | | | | | | |
| Nuclear | 445 (31.5) | 368 (49.9) | 77 (11.4) | 542 (38.4) | 395 (53.5) | 147 (21.8) | 944 (66.9) | 616 (83.5) | 328 (48.7) | 1093 (77.5) | 656 (88.9) | 437 (64.9) | 462 (32.7) | 378 (51.2) | 84 (12.5) |
| Extended | 159 (30.7) | 105 (47.3) | 54 (18.2) | 200 (38.6) | 116 (52.3) | 84 (28.4) | 348 (67.2) | 176 (79.3) | 172 (58.1) | 411 (79.3) | 195 (87.8) | 216 (73.0) | 162 (31.3) | 105 (47.3) | 57 (19.3) |

[1] Showing the percentages of people within a category (such as men with no childhood emotional abuse) who refused wife abuse (comparing to the men who did not refuse the abuse).

Table 3. Bivariate binary logistic regressions presenting the unadjusted ORs for different types of attitudinal refusal of wife abuse, N = 1929.

| | Refused overall wife abuse | | Refused emotional abuse | | Refused physical abuse | | Refused abuse on disobeying obligations | | Refused abuse on challenging authority | |
|---|---|---|---|---|---|---|---|---|---|---|
| | OR (95% CI) | P | OR (95% CI) | P | OR (95% CI) | P | OR (95% CI) | P | OR (95% CI) | P |
| **Childhood emotional abuse** | | | | | | | | | | |
| None | 8.75 (5.47–13.98) | <0.001 | 2.83 (2.02–3.98) | <0.001 | 3.83 (2.74–5.36) | <0.001 | 2.92 (2.01–4.25) | <0.001 | 8.15 (5.20–12.79) | <0.001 |
| Mild | 5.32 (3.45–8.21) | <0.001 | 1.76 (1.32–2.35) | <0.001 | 3.46 (2.66–4.51) | <0.001 | 2.25 (1.72–2.97) | <0.001 | 4.87 (3.22–7.36) | <0.001 |
| Severe | 1 | | 1 | | 1 | | 1 | | 1 | |
| **Childhood physical Abuse** | | | | | | | | | | |
| None | 2.68 (2.01–3.44) | <0.001 | 2.08 (1.62–2.68) | <0.001 | 2.71 (2.09–3.52) | <0.001 | 2.46 (1.85–3.27) | <0.001 | 2.63 (2.01–3.44) | <0.001 |
| Mild | 1.36 (1.03–1.74) | 0.026 | 1.35 (1.05–1.72) | 0.017 | 1.64 (1.29–2.07) | <0.001 | 1.86 (1.44–2.41) | <0.001 | 1.34 (1.03–1.74) | 0.031 |
| Severe | 1 | | 1 | | 1 | | 1 | | 1 | |
| **Ethnicity** | | | | | | | | | | |
| Garo | 1.90 (1.49–2.43) | <0.001 | 1.34 (1.07–1.68) | 0.011 | 1.88 (1.49–2.38) | <0.001 | 0.84 (0.64–1.11) | 0.215 | 1.96 (1.54–2.49) | <0.001 |
| Santal | 1.76 (1.37–2.24) | <0.001 | 1.24 (0.99–1.56) | 0.062 | 1.46 (1.16–1.83) | 0.001 | 0.63 (0.48–0.82) | 0.001 | 1.86 (1.46–2.37) | <0.001 |
| Bengali | 1 | | 1 | | 1 | | 1 | | 1 | |
| **Gender** | | | | | | | | | | |
| Men | 6.21 (4.96–7.77) | <0.001 | 3.64 (2.99–4.42) | <0.001 | 4.42 (3.59–5.45) | <0.001 | 3.78 (2.97–4.80) | <0.001 | 5.95 (4.78–7.40) | <0.001 |
| Women | 1 | | 1 | | 1 | | 1 | | 1 | |
| **Age in years** | | | | | | | | | | |
| 16–25 | 0.33 (0.22–0.50) | <0.001 | 0.45 (0.31–0.67) | <0.001 | 0.58 (0.39–0.87) | 0.008 | 0.76 (0.47–1.22) | 0.253 | 0.31 (0.20–0.46) | <0.001 |
| 26–35 | 0.61 (0.44–0.87) | 0.005 | 0.66 (0.47–0.93) | 0.017 | 0.92 (0.64–1.31) | 0.642 | 0.72 (0.47–1.09) | 0.124 | 0.57 (0.40–0.79) | 0.001 |
| 36–45 | 0.81 (0.57–1.13) | 0.218 | 0.83 (0.59–1.17) | 0.292 | 1.09 (0.76–1.56) | 0.655 | 0.83 (0.54–1.27) | 0.389 | 0.75 (0.53–1.05) | 0.091 |
| 46–60 | 1 | | 1 | | 1 | | 1 | | 1 | |
| **Schooling** | | | | | | | | | | |
| Higher | 2.08 (1.34–3.22) | 0.001 | 1.11 (0.76–1.63) | 0.587 | 2.56 (1.73–3.79) | <0.001 | 2.33 (1.48–3.67) | <0.001 | 2.07 (1.34–3.19) | 0.001 |
| Secondary | 1.39 (0.91–2.13) | 0.130 | 0.83 (0.58–1.21) | 0.336 | 1.98 (1.38–2.86) | <0.001 | 1.52 (1.01–2.29) | 0.043 | 1.39 (0.91–2.13) | 0.122 |
| Primary | 1.68 (1.11–2.54) | 0.014 | 0.90 (0.63–1.29) | 0.569 | 2.07 (1.45–2.94) | <0.001 | 1.22 (0.83–1.81) | 0.306 | 1.72 (1.14–2.58) | 0.010 |
| None | 1 | | 1 | | 1 | | 1 | | 1 | |
| **Monthly income** | | | | | | | | | | |
| BDT 7000/above | 3.89 (2.89–5.25) | <0.001 | 2.03 (1.58–2.62) | <0.001 | 4.14 (3.17–5.40) | <0.001 | 2.47 (1.82–3.35) | <0.001 | 3.78 (2.82–5.08) | <0.001 |
| Below BDT 7000 | 2.13 (1.58–2.88) | <0.001 | 1.28 (0.99–1.65) | 0.055 | 1.63 (1.28–2.07) | <0.001 | 0.94 (0.72–1.22) | 0.641 | 2.19 (1.63–2.95) | <0.001 |
| No income | 1 | | 1 | | 1 | | 1 | | 1 | |
| **Family type** | | | | | | | | | | |
| Nuclear | 1.04 (0.84–1.29) | 0.723 | 0.99 (0.81–1.22) | 0.937 | 0.99 (0.79–1.22) | 0.908 | 0.89 (0.69–1.14) | 0.377 | 1.07 (0.86–1.33) | 0.541 |
| Extended | 1 | | 1 | | 1 | | 1 | | 1 | |

**Table 4. Multivariate binary logistic regressions presenting the adjusted ORs for different types of attitudinal refusal of wife abuse, $N = 1929$.**

| | Refused overall wife abuse | | Refused emotional abuse | | Refused physical abuse | | Refused abuse on disobeying obligations | | Refused abuse on challenging authority | |
|---|---|---|---|---|---|---|---|---|---|---|
| | OR (95% CI) | P | OR (95% CI) | P | OR (95% CI) | P | OR (95% CI) | P | OR (95% CI) | P |
| **Childhood emotional abuse** | | | | | | | | | | |
| None | 1.75 (1.02–3.01) | 0.041 | 0.81 (0.53–1.23) | 0.323 | 0.93 (0.60–1.43) | 0.742 | 1.05 (0.66–1.69) | 0.827 | 1.63 (0.97–2.75) | 0.064 |
| Mild | 2.31 (1.44–3.69) | <0.001 | 0.95 (0.69–1.32) | 0.773 | 1.78 (1.33–2.39) | <0.001 | 1.41 (1.03–1.93) | 0.032 | 2.12 (1.35–3.31) | 0.001 |
| Severe | 1 | | 1 | | 1 | | 1 | | 1 | |
| **Childhood physical Abuse** | | | | | | | | | | |
| None | 1.84 (1.36–2.50) | 0.031 | 1.66 (1.25–2.19) | <0.001 | 2.05 (1.53–2.75) | <0.001 | 1.79 (1.30–2.46) | <0.001 | 1.83 (1.35–2.47) | <0.001 |
| Mild | 1.29 (0.96–1.73) | 0.465 | 1.34 (1.03–1.74) | 0.028 | 1.62 (1.26–2.10) | <0.001 | 1.68 (1.28–2.22) | <0.001 | 1.28 (0.95–1.71) | 0.098 |
| Severe | 1 | | 1 | | 1 | | 1 | | 1 | |
| **Ethnicity** | | | | | | | | | | |
| Garo | 2.24 (1.69–2.97) | <0.001 | 1.62 (1.25–2.09) | <0.001 | 2.19 (1.66–2.90) | <0.001 | 0.88 (0.64–1.22) | 0.447 | 2.28 (1.72–3.02) | <0.001 |
| Santal | 2.00 (1.50–2.67) | <0.001 | 1.28 (0.99–1.65) | 0.057 | 1.82 (1.39–2.37) | <0.001 | 0.69 (0.51–0.93) | 0.013 | 2.12 (1.59–2.82) | <0.001 |
| Bengali | 1 | | 1 | | 1 | | 1 | | 1 | |
| **Gender** | | | | | | | | | | |
| Men | 8.55 (5.99–12.23) | <0.001 | 6.18 (4.49–8.51) | <0.001 | 5.10 (3.67–7.09) | <0.001 | 4.86 (3.41–6.92) | <0.001 | 7.94 (5.60–11.25) | <0.001 |
| Women | 1 | | 1 | | 1 | | 1 | | 1 | |
| **Age in years** | | | | | | | | | | |
| 16–25 | 1.27 (0.77–2.12) | 0.350 | 1.19 (0.75–1.89) | 0.458 | 2.14 (1.32–3.48) | 0.002 | 2.06 (1.18–3.58) | 0.011 | 1.11 (0.67–1.83) | 0.685 |
| 26–35 | 1.40 (0.95–2.08) | 0.092 | 1.27 (0.87–1.86) | 0.212 | 2.42 (1.58–3.69) | <0.001 | 1.46 (0.90–2.38) | 0.122 | 1.25 (0.84–1.84) | 0.271 |
| 36–45 | 1.19 (0.82–1.74) | 0.345 | 1.10 (0.77–1.58) | 0.595 | 1.88 (1.25–2.82) | 0.002 | 1.10 (0.69–1.76) | 0.675 | 1.09 (0.75–1.58) | 0.644 |
| 46–60 | 1 | | 1 | | 1 | | 1 | | 1 | |
| **Schooling** | | | | | | | | | | |
| Higher | 1.99 (1.19–3.30) | 0.008 | 0.89 (0.58–1.37) | 0.605 | 2.07 (1.32–3.23) | 0.001 | 1.53 (0.93–2.52) | 0.095 | 2.06 (1.25–3.41) | 0.005 |
| Secondary | 0.99 (0.61–1.63) | 0.989 | 0.58 (0.39–0.88) | 0.011 | 1.46 (0.96–2.22) | 0.076 | 0.91 (0.58–1.43) | 0.684 | 1.05 (0.65–1.70) | 0.845 |
| Primary | 1.08 (0.68–1.73) | 0.736 | 0.60 (0.41–0.90) | 0.012 | 1.35 (0.91–2.01) | 0.136 | 0.73 (0.48–1.12) | 0.150 | 1.15 (0.72–1.82) | 0.559 |
| None | 1 | | 1 | | 1 | | 1 | | 1 | |
| **Monthly income** | | | | | | | | | | |
| BDT 7000/above | 0.51 (0.33–0.80) | 0.003 | 0.42 (0.29–0.61) | <0.001 | 1.09 (0.75–1.58) | 0.649 | 0.90 (0.60–1.36) | 0.628 | 0.52 (0.34–0.81) | 0.004 |
| Below BDT 7000 | 0.61 (0.41–0.90) | 0.013 | 0.55 (0.40–0.75) | <0.001 | 0.86 (0.65–1.15) | 0.307 | 0.70 (0.52–0.95) | 0.023 | 0.65 (0.44–0.95) | 0.027 |
| No income | 1 | | 1 | | 1 | | 1 | | 1 | |
| **Model summary** | | | | | | | | | | |
| Omnibus χ2 (15, $N = 1929$) | 407.41 | <0.001 | 238.30 | <0.001 | 345.714 | <0.001 | 219.33 | <0.001 | 402.17 | <0.001 |
| −Log-Likelihood | 1990.61 | | 2332.27 | | 2101.55 | | 1815.05 | | 2026.31 | |
| Nagelkerke R Square | 0.27 | | 0.16 | | 0.23 | | 0.17 | | 0.26 | |

**Refusal of abuse on wife disobeying family obligations.** Table 4 shows that the ORs for the refusal of abuse on disobeying family obligations was 1.05 (*p* = .827) for the non-victims of childhood emotional abuse and 1.41 (*p* = .032) for the victims of mild emotional abuse compared to the victims of severe emotional abuse. It further reveals that the ORs for the refusal of wife abuse on disobeying family obligations was 1.79 (*p* < .001) for the non-victims of childhood physical abuse, and 1.68 (*p* < .001) for the victims of mild physical abuse, compared to the childhood victims of severe physical abuse.

**Refusal of abuse on wife challenging male authority.** Data show that the ORs for the refusal of wife abuse on challenging male authority was 1.63 (*p* = .064) for the non-victims of childhood emotional abuse and 2.12 (*p* = .001) for the victims of mild emotional abuse compared to the victims of severe emotional abuse. It further reveals that the ORs for the refusal of wife abuse on challenging male authority was 1.83 (*p* < .001) for the non-victims of childhood physical abuse, and 1.28 (*p* = .098) for the victims of mild abuse, compared to the victims of severe physical abuse (see Table 4).

Furthermore, the study shows that the ORs for the refusal of both emotional and physical abuses as well as the refusal of abuse on challenging male authority were higher among the *Garo* and *Santal* than the *Bengali* communities. However, the refusal of abuse on wife's disobeying family obligations appeared to be less among the *Santal* than the *Bengali*. People with higher education were also more likely to refuse overall wife abuse, physical abuse, and abuse on challenging male authority. All forms of attitudinal refusal of wife abuse also appeared to be higher among the men than the women (see Table 4). From the gender-disaggregated analysis, our study further reveals that women's attitude against wife abuse was influenced mainly by their childhood experience of emotional abuse. In contrast, men's attitude was shaped by their childhood experience of physical abuse (see S2 and S3 Tables).

Though the refusal of wife abuse was usually higher among the ethnic minority *Garo* and *Santal* communities than the mainstream *Bengali* community, we also observed gender differences in associations between outcomes and ethnicity (see S2 and S3 Tables). The refusal of overall wife abuse, refusal of emotional wife abuse, and refusal of wife abuse on challenging male authority appeared to be higher among Garo and Santal men than the Bengali men. However, there were no significant differences in the attitudinal refusal of physical wife abuse and the refusal of wife abuse on disobeying family obligations between the male respondents of different communities. On the other hand, women from both Garo and Santal communities attitudinally refused more overall wife abuse, physical abuse, and abuse on challenging male authority than the women from the Bengali community. At the same time, there were no significant differences in the refusal of emotional abuse between the women from different ethnicities. Conversely, the refusal of wife abuse on disobeying family obligations appeared to be lower among Garo and Santal women than the Bengali women.

## Discussion

The study shows that the childhood experience of family victimization has a great influence on the adulthood attitudinal refusal of wife abuse in rural Bangladesh. It appears that the refusals of overall wife abuse, emotional abuse, physical abuse, abuse on disobeying family obligations, and abuse on challenging male family authority are higher among the childhood non-victims of physical abuse than those exposed to severe physical abuse during childhood. Compared to severe emotional abuse, the study also indicates that childhood exposure to mild emotional abuse may increase the refusal of overall wife abuse, physical abuse, abuse on disobeying family obligations, and abuse on challenging male authority.

The findings of our study are consistent with a body of previous studies indicating that childhood experiences of family abuse are influential on the refusal/acceptance of wife abuse in adulthood [7, 45–48]. The preceding studies primarily examined the factors influencing attitudes toward wife beating [11, 47–49]. In contrast, our study estimated the refusal of wife abuse on its five dimensions: refusal of overall wife abuse, refusal of emotional abuse, refusal of physical abuse, refusal of abuse on disobeying family obligations, and refusal of abuse on challenging male authority. In order to examine the effects of childhood family victimization, most of the previous studies considered childhood familial victimization as a single category; moreover, they often included the witnessing of inter-parental abusive behaviors [5, 11, 46–48, 50]. However, our current study systematically examined the implications of childhood exposure to both emotional and physical abuses in the family of origin. We also estimated the severity of such childhood abuses. Our study further indicates that not all types of childhood victimization (or non-victimization) equally influence the adulthood refusal of wife abuse. For example, our data suggest that childhood non-exposure to physical abuse is the most crucial determinant in increasing refusal of wife abuse in rural Bangladesh. Childhood exposure to mild physical abuse also appears to be significantly influential in the refusal of different forms of wife abuse than childhood exposure to severe physical abuse. On the other hand, childhood exposure to mild emotional abuse within the family of origin appears to increase the adulthood attitudinal refusal of wife abuse, compared to childhood exposure to severe emotional abuse. These results provide a novel understanding of the issue and add to the current literature. It shows that not all types of abusive childhood experiences contribute equally to the adulthood attitudinal refusal of wife abuse. We also observed that women's attitudes toward wife abuse were mainly influenced by their childhood experience of emotional abuse, while men's attitudes were shaped by their experience of physical abuse.

Some of these findings are, to some extent, inconsistent with the notion of learning theory [24–26, 51] and a large body of previous empirical studies [e.g., 6, 7, 11, 25, 46–49, 52]. Regarding the childhood victimization of emotional abuse, our study indicates that those who experienced mild emotional abuse during childhood are more likely to refuse the wife abuse in adulthood than those who experienced severe emotional abuse. These findings seem to be difficult to explain, but we speculate that children may perceive mild emotional parental abuse as a form of discipline rather than abuse. It is plausible that children and parents did not consider mild emotional abuse as an 'abusive act.' Even though the act is abusive, they might view it as 'normal' and a form of discipline. Thus, the learning of abuse from childhood experiences of mild emotional abuse might differ from other abusive acts like severe emotional and physical abuses. The field notes of our interviewers indicated that most of the respondents commented on mild emotional abuse as quite a 'normal act.' They believed that their parents used this form of discipline to correct their behaviors.

The study further shows that men are more likely to refuse all types of wife abuse than women in rural Bangladesh. These results are in line with few previous studies showing that women uphold more conservative gender ideology than their male counterparts in low-income patriarchal societies like Bangladesh [7, 11, 12]. This is possible because both men and women can accept, maintain and reinforce the patriarchal social order. As a 'subordinated gender,' women's acceptance/reinforcement of these patriarchal traditions makes sure the continuation of wife abuse/male domination over women [12, 53]. This indicates that most of the women are not so aware of their rights.

Our findings also reveal that respondents from the *Garo* and *Santal* ethnic minority communities are more likely to refuse most types of wife abuse than the mainstream *Bengali* communality. Although a previous study [12] indicated similar findings, our current study further shows that these ethnic minority communities did not refuse wife abuse on the ground of the

wife's disobeying family obligations more than the mainstream *Bengali* community. It also reveals that there are gender differences in the associations between outcomes and ethnicity. The feature of the refusal of physical wife abuse and the refusal of wife abuse on disobeying family obligations appears similar among the male respondents of different communities. On the other hand, women of different ethnicities do not differ from each other in refusing emotional wife abuse. Conversely, the refusal of wife abuse on disobeying family obligations appears lower among Garo and Santal women than the Bengali women.

Our study further shows that people with higher education are more likely to refuse few types of wife abuse. These findings are consistent, to some extent, with those of other studies [12, 53], arguing that education may generally increase the refusal of wife abuse. However, our study shows that higher education was associated with an increased refusal of two critical types of wife abuse: physical abuse and abuse on challenging male authority. In contrast, education did not appear to influence the refusal of emotional abuse and the refusal of abuse on wife disobeying family obligations. We believe that these two aspects of wife abuse may be learned/internalized by most social members during their childhood socialization, but this learning may be further rationalized in adulthood. The low intensity and low social costs associated with these types of abuse might be related to their acceptance of wife abuse. On the other hand, regarding physical abuse and abuse on challenging male authority, we speculate that people with higher education might be de-motivated to engage in these behaviors in adulthood. People with higher education may become more sensible/understanding, uphold liberal gender ideology, and be more aware of the consequences of abuse, and therefore, more likely to refuse these types of wife abuse. We believe that education is the key to altering childhood learning of these negative behaviors.

Broadly speaking, the findings of our study support the propositions of the social learning theory of wife abuse, which argues that wife abuse is a socially learned behavior and that children observe and internalize abusive behaviors through childhood experiences [21, 24–26, 35, 51]. The current study supports the idea that the family of origin is the training ground wherein children learn when and under what conditions he/she could refuse or accept wife abuse [21, 25]. If the children do not experience abuse in the family, they are more likely to refuse the abuse in adulthood. Our findings notably indicate that it may be possible to alter childhood learning of some abuses in adulthood with proper education.

## Limitations and future directions

The present study has several limitations. Due to budget constraints, it was not possible to adopt a prospective cohort design. Future studies could adopt a longitudinal design and adopt a prospective cohort design in order to follow the effects of exposure variables on the refusal of wife abuse. The current study lacks the explanation of why childhood exposure to mild emotional abuse may increase the refusal of wife abuse in rural Bangladesh. It was expected that the attitudinal refusal of wife abuse would be higher among the matrilineal communities than the patrilineal communities. However, we had a surprising finding, and we lack a proper explanation of why women from ethnic communities refused less abuse on the ground of the wife's disobeying family obligations than mainstream Bengali women. An explorative qualitative study may answer these research questions thoroughly.

## Conclusions

The attitudinal refusal of wife abuse is quite low in rural Bangladesh. People's exposure to different types of familial victimization during childhood is also widespread. These findings may explain why wife abuse is so pervasive in society. Our study suggests that childhood family

non-victimization may increase the adulthood attitudinal refusal of wife abuse. This denotes that it is crucial to initiate appropriate child welfare programs to increase the attitudinal refusal of wife abuse. The current study highlights that the prevention of child abuse at the parental home may reduce the pervasive wife abuse in the family of procreation. It shows that childhood experience of family victimization has a significant influence on the attitudinal refusal/ acceptance of wife abuse in rural Bangladesh. We argue that the issue of childhood abuse should be brought to the forefront in the discourse of wife abuse.

## Implications and recommendations

We observe that childhood family victimization has a great influence on the refusal (or acceptance) of wife abuse in rural Bangladesh. The study indicates that those who have not experienced abuse in their family during childhood are more likely to refuse wife abuse. The study reveals that childhood experiences of different types of familial abuse have varying influences on the different domains of abuse–where people accept/refuse wife abuse. These are important findings that might be used to enhance the attitudinal refusal of wife abuse in rural Bangladesh. The findings may help policy makers, researchers, clinicians, and other practitioners to address widespread wife abuse in the country. It may stimulate them to think about strategies for the primary prevention of wife abuse from more constructive viewpoints. We recommend that the state machinery and social welfare agencies should expend significant efforts to stop child abuse within the family and in society generally. It is also essential to have appropriate educational programs (including community education) so that individuals get de-motivated to resort to any type of violence against women in adulthood.

## Supporting information

**S1 Table. Items used in the outcome and main exposure variables.**
(PDF)

**S2 Table. Multivariate logistic regression models for women.**
(PDF)

**S3 Table. Multivariate logistic regression models for men.**
(PDF)

**S1 File. Dataset.**
(SAV)

## Author Contributions

**Conceptualization:** Rabiul Karim, Katarina Swahnberg.

**Data curation:** Rabiul Karim, Suchona Rahman, Hafijur Rahman.

**Formal analysis:** Rabiul Karim.

**Funding acquisition:** Rabiul Karim, Katarina Swahnberg.

**Investigation:** Rabiul Karim, Suchona Rahman, Hafijur Rahman, Sadequl Arefin, Katarina Swahnberg.

**Methodology:** Rabiul Karim, Katarina Swahnberg.

**Project administration:** Rabiul Karim, Tanzima Zohra Habib, Sadequl Arefin, Katarina Swahnberg.

**Resources:** Rabiul Karim, Katarina Swahnberg.

**Software:** Rabiul Karim.

**Supervision:** Rabiul Karim, Katarina Swahnberg.

**Validation:** Rabiul Karim, Suchona Rahman, Hafijur Rahman, Katarina Swahnberg.

**Visualization:** Rabiul Karim.

**Writing – original draft:** Rabiul Karim.

**Writing – review & editing:** Suchona Rahman, Tanzima Zohra Habib, Katarina Swahnberg.

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
