## [Decision Letter · Decision Letter 0]

25 Nov 2020

PONE-D-20-30007

Does Childhood Experience of Family Victimization influence Adulthood Refusal of Wife Abuse? Evidence from Rural Bangladesh

PLOS ONE

Dear Dr. Karim,

Thank you for submitting your manuscript to PLOS ONE. After careful consideration, we feel that it has merit but does not fully meet PLOS ONE’s publication criteria as it currently stands. Therefore, we invite you to submit a revised version of the manuscript that addresses the points raised during the review process.

Both the reviewers have made substantive suggestions to help strengthen the manuscript. To help guide you through these revisions, please use the STROBE guidelines and attach a completed checklist with your revised manuscript. That will help you as authors, myself as editor, and our two reviewers to track how the manuscript reflects best practices in reporting (see https://www.strobe-statement.org/?id=available-checklists). 

We look forward to receiving your revised manuscript.

Kind regards,

Kristin Dunkle

Academic Editor

PLOS ONE

Journal Requirements:

2. For studies involving humans categorized by race/ethnicity, age, disease/disabilities, religion, sex/gender, sexual orientation, or other socially constructed groupings, authors should: 1) Explicitly describe their methods of categorizing human populations, 2) Define categories in as much detail as the study protocol allows, 3) Justify their choices of definitions and categories, 4) Explain whether (and if so, how) they controlled for confounding variables such as socioeconomic status, nutrition, environmental exposures, or similar factors in their analysis.

4.Thank you for submitting the above manuscript to PLOS ONE. During our internal evaluation of the manuscript, we found some occurrences of overlapping text with the following previous publication(s), some of which you are an author, which needs to be addressed:

- https://journals.plos.org/plosone/article?id=10.1371/journal.pone.0236733

Please revise the manuscript to quote or rephrase the duplicated text and cite your sources for text outside the methods section. Please note that further consideration is dependent on the submission of a manuscript that addresses these concerns about the overlap in text with published work.

Reviewers' comments:

Reviewer's Responses to Questions

**Comments to the Author**

1. Is the manuscript technically sound, and do the data support the conclusions?

Reviewer #1: Partly

Reviewer #2: Yes

2. Has the statistical analysis been performed appropriately and rigorously? 

Reviewer #1: No

Reviewer #2: Yes

3. Have the authors made all data underlying the findings in their manuscript fully available?

Reviewer #1: No

Reviewer #2: Yes

4. Is the manuscript presented in an intelligible fashion and written in standard English?

Reviewer #1: No

Reviewer #2: Yes

5. Review Comments to the Author

Reviewer #1: The authors have tackled an important public health problem that is crucial in primary prevention of intimate partner violence. However, I have several pertinent issues of concern with the study/paper, which I will outline below.

i) One of the main concern is on how the authors have defined the 2 main exposures ( having multiple dichotomous variables that make it difficult to interpret their multivariate model results. In their description of the exposures they have defined the following categories “mild”, “moderate”, “severe” and “no victimisation”, which are assumed to be mutually exclusive categories. However, having multiple dichotomous variables (e.g moderate vs no victimization) as shown in Table 1 & 2 is not an efficient way of looking at the exposure. As Table 1 shows, participants classified as ‘mild’ include ‘moderate’ and ‘severe’ categories.

I would recommend that the authors define the exposures (emotional victimization & physical victimization) using 2 variables that have mutually exclusive categories (no victimization, mild, moderate and severe).

ii) The authors should include a Table showing items used to measure the outcomes and also the main exposures. This would make it easier for the reader to understand how these have been measured and defined. As it is now, it is rather difficult to understand how the outcomes and exposures have been derived.

iii) The authors need to be consistent in how they refer to the main exposures. As it is, they talk of childhood victimization (which is a risk factor) and then switch to childhood non-victimization (as a protective factors). Most literature tend to look at childhood victimization as a being positively associated with acceptance of wife abuse, which is basically conversely same as looking at childhood non-victimization being positively associated with refusal of wife abuse (as defined by the authors). They need to be consistent in how they present these in the paper and Tables. Otherwise moving from one to the other is rather confusing. They may find that it is easier to look at victimization & acceptance attitude (going by the way they have presented their discussion and abstract).

iv) I understand that their paper is part of a bigger study which has both men and women. It would be interesting to see how their results differ (or not differ) if the authors performed subgroup analysis (by gender), especially the prevalence of ‘refusal of wife abuse.

v) Other general comments that the authors need address are:

• They need to be a bit more succinct. They do not need to have a background & introduction. They need to cut down on the methods section especially their description of the Site of the study.

• It is not clear if the sample size calculation presented in the paper is for this particular study or is for the main C-RCT. If it is for the main C-RCT, they need to make it clear. Otherwise it is difficult to link the parameters in their formula to the study objectives of this paper.

Reviewer #2: Thank you for this paper which deals with a very critical topic, below I offer some feedback on the paper, I hope it is useful in strengthening the paper.

Abstract

I would suggest being clear from the beginning, and in the abstract who you’re referring to (men only or men and women) when you speak about “attitudinal refusal of wife abuse” I suggest saying ‘attitudinal refusal by women and men, of wife abuse”. Indeed, this needs to be clarified throughout, and I suggest later you may want to analyse and discuss by gender as well.

You indicate the % that “refused different types of wife abuse” – is this based on their views/attitudes, ie: do you think x is acceptable or not etc. Or is it based on whether the person had perpetrated any of these acts, ie: did you ask “have you ever done x” or “have you done x in the last 12 months” etc?? Please clarify for the reader. I am assuming as you refer to ARWA that this is ‘attitudes’ not actual actions. Please clarify this throughout the paper.

“We argue that the issues of childhood victimization should be brought to the center in the discourse” Please could you make this a much stronger recommendation, who would you call upon to do this and where? For example are you calling for more research, changes in policy???? And where? In Bangladesh or where???

Introduction

As I stressed in my comments above please begin by indicating who you are speaking about “Attitudinal refusal of wife abuse (ARWA)…” – by whom? Men only or men and women.

Theory:

You make the following point “though little is known about why many adults who were exposed to childhood abusive events do not continue the same pattern with their partners” this is a very important point and I think it would be really good if you could add some reflections on what the literature is saying about this, even if there is not much. Then in the discussion you could see if your results and analysis are similar in anyway etc.

Methods:

The methods section could be reduced quite a bit.

Study participants, you state clearly it was adults under 60, please indicate the youngest age for inclusion.

“We excluded men/women with mental/physical disabilities.” Please indicate why they were excluded.

‘Social Acceptance of Wife Abuse Scale’ (SAWAS) – its great that you used a locally developed and validated scale.

The childhood exposure questions – please specify whether these covered both experience of and witnessing abuse (I assume it did?), and if it includes witnessing as well, perhaps include an example of one.

Results

I think it is important to discuss the results by men and women’s experiences – I think it is important to understand how these views and experiences differ. It would also be interesting to discuss the analysis by age, which would show, among other things, if there are any generational shifts occurring etc. Please can you undertake and present these results as well. Its interesting to see the effect of education and ethnicity and I imagine gender and age would also reveal interesting findings.

Discussion

I feel that discussions should not try to present all the data again, and should make a relatively smooth, yet powerful, read, so I think you might want to re-write the first paragraph, could you shorten it to make it more readable. The final sentence in the first paragraph does a great job of summarizing some results very effectively, perhaps you could consider something like that for the beginning of the paragraph

“These findings have offered a novel understanding of the issue and added to the current literatures that not all childhood abusive experiences equally contribute to the adulthood attitudinal refusal/acceptance of wife abuse” This is an important finding, and if we could also have this broken down by gender and age we would learn so much more!

Please consider a final edit of the paper as there are still a few grammatical errors.

6. PLOS authors have the option to publish the peer review history of their article (what does this mean?). If published, this will include your full peer review and any attached files.

Reviewer #1: No

Reviewer #2: No

---

## [Author Response · Author response to Decision Letter 0]

5 Jan 2021

Reviewer #1: 

Comment-1:

The authors have tackled an important public health problem that is crucial in primary prevention of intimate partner violence. However, I have several pertinent issues of concern with the study/paper, which I will outline below.

i) One of the main concern is on how the authors have defined the 2 main exposures (having multiple dichotomous variables that make it difficult to interpret their multivariate model results. In their description of the exposures they have defined the following categories “mild”, “moderate”, “severe” and “no victimisation”, which are assumed to be mutually exclusive categories. However, having multiple dichotomous variables (e.g moderate vs no victimization) as shown in Table 1 & 2 is not an efficient way of looking at the exposure. As Table 1 shows, participants classified as ‘mild’ include ‘moderate’ and ‘severe’ categories.

I would recommend that the authors define the exposures (emotional victimization & physical victimization) using 2 variables that have mutually exclusive categories (no victimization, mild, moderate and severe).

Response-1: Thank you for the very useful comments. We have re-categorized both the exposure variables into: ‘none,’ ‘mild,’ and ‘severe.’ Based on the similar pattern of occurrence, we merged mild/moderate emotional abuses and defined them as ‘mild.’ On the other hand, we merged moderate/severe physical abuses and defined them as ‘severe.’ Both the exposure variables (childhood emotional abuse and childhood physical abuse) are now well defined and easy to interpret. The changes are reflected in different paragraphs and tables, for example, pages 9-10, lines 215-235; page 14, line 319 (table 2); page 15, line 340 (table 3); and page 18, line 391 (table 4).

Comment-2:

ii) The authors should include a Table showing items used to measure the outcomes and also the main exposures. This would make it easier for the reader to understand how these have been measured and defined. As it is now, it is rather difficult to understand how the outcomes and exposures have been derived.

Response-2: Unfortunately, this will increase the length of the paper and the number of tables. We already have four tables. The items are discussed in defining the variables (see pages 8-10, lines 191-235). However, we included a table on the items used in the outcome and main exposure variables, respectively as supporting information file (see S1_Table). We also referred to them in the description of the study measures: page 8, line 195; and page 9, line 221. 

Comment-3:

iii) The authors need to be consistent in how they refer to the main exposures. As it is, they talk of childhood victimization (which is a risk factor) and then switch to childhood non-victimization (as a protective factors). Most literature tend to look at childhood victimization as a being positively associated with acceptance of wife abuse, which is basically conversely same as looking at childhood non-victimization being positively associated with refusal of wife abuse (as defined by the authors). They need to be consistent in how they present these in the paper and Tables. Otherwise moving from one to the other is rather confusing. They may find that it is easier to look at victimization & acceptance attitude (going by the way they have presented their discussion and abstract).

Response-3: Thank you. We have tried to be consistent and revised the paper accordingly. 

Comment-4:

iv) I understand that their paper is part of a bigger study which has both men and women. It would be interesting to see how their results differ (or not differ) if the authors performed subgroup analysis (by gender), especially the prevalence of ‘refusal of wife abuse.

Response-4: With this revision, we have now presented gender-disaggregated analysis/findings. For the refusal of wife abuse, see page 13, lines 297-302; page 13, line 304 (table 1). For childhood exposure to familial abuse, see page 13, lines 306-313; page 14, line 319 (table 2). Gender is also used as a controlled variable (moderator) in the final models: page 17, lines 387-390; page 18, line 391 (table 4). We also conducted subgroup analysis by gender and found that women are generally influenced by childhood emotional abuse, while men are mostly influenced by physical abuse: page 17, lines 387-390. We included these findings of multivariate analysis as supporting files: ‘S2_Table’ for women; ‘S3_Table’ for men.

Comment-5:

v) Other general comments that the authors need address are:

• They need to be a bit more succinct. They do not need to have a background & introduction. They need to cut down on the methods section especially their description of the Site of the study.

Response-5: We have now shortened the sections. We merged the introduction and the background and shortened it further. We also shortened the method section and description of study sites.

Comment-6:

• It is not clear if the sample size calculation presented in the paper is for this particular study or is for the main C-RCT. If it is for the main C-RCT, they need to make it clear. Otherwise it is difficult to link the parameters in their formula to the study objectives of this paper.

Response-6: Based on the findings of our pilot study, the sample size was estimated for the main C-RCT study, and this is now made more clear in the text. Please see page 8, lines 179-180. 

Reviewer #2: 

Comment-7:

Thank you for this paper which deals with a very critical topic, below I offer some feedback on the paper, I hope it is useful in strengthening the paper.

Abstract

I would suggest being clear from the beginning, and in the abstract who you’re referring to (men only or men and women) when you speak about “attitudinal refusal of wife abuse” I suggest saying ‘attitudinal refusal by women and men, of wife abuse”. Indeed, this needs to be clarified throughout, and I suggest later you may want to analyse and discuss by gender as well.

Response-7: We have now made it (i.e., who we are referring to - men only or men and women) clear in the abstract as well as throughout the paper. Now we clearly use ‘attitudinal refusal of wife abuse among men and women.’ In the paper, we analyzed the data by gender as well as tried to discuss the findings accordingly. We made the suggested changes in the abstract: page 2, lines 31-36. The results related to the refusal of wife abuse also now embraced with gender disaggregated findings: page 13, lines 297-302; page 13, line 304 (table 1); the exposure variable (childhood victimization) also included findings by gender: page 13, lines 306-313; page 14, line 319 (table 2).

Gender is also used as a controlled variable in the final models: page 17, lines 387-390; page 18, line 391 (table 4). 

With this revision, we have also conducted subgroup analysis by gender and found that women are generally influenced by childhood emotional abuse, while men are mostly influenced by physical abuse: page 317, lines 387-390. We included these findings of multivariate analysis as supporting files: ‘S2_Table’ for women; ‘S3_Table’ for men.

Comment-8:

You indicate the % that “refused different types of wife abuse” – is this based on their views/attitudes, ie: do you think x is acceptable or not etc. Or is it based on whether the person had perpetrated any of these acts, ie: did you ask “have you ever done x” or “have you done x in the last 12 months” etc?? Please clarify for the reader. I am assuming as you refer to ARWA that this is ‘attitudes’ not actual actions. Please clarify this throughout the paper.

Response-8: Thank you for pointing this out. In this study, attitudinal refusal of wife abuse means a non-violent attitude toward wife abuse, not the actual action. We have now made this clear, both in the abstract and in the introduction: page 2, line 31; page 3, line 55. 

Comment-9:

“We argue that the issues of childhood victimization should be brought to the center in the discourse” Please could you make this a much stronger recommendation, who would you call upon to do this and where? For example are you calling for more research, changes in policy???? And where? In Bangladesh or where???

Response-9: We have now made a more stronger recommendation in the abstract, page 2, lines 50-52, and in the discussion section: page 21, line 463-465; page 22, line 487-495.

Comment-10:

Introduction

As I stressed in my comments above please begin by indicating who you are speaking about “Attitudinal refusal of wife abuse (ARWA)…” – by whom? Men only or men and women.

Response-10: Please see our response no.8 

We have now made it clear in the beginning of the paragraph (introduction). We use ‘attitudinal refusal of wife abuse among men and women’; please see page 3, line 55.

Comment-11:

Theory:

You make the following point “though little is known about why many adults who were exposed to childhood abusive events do not continue the same pattern with their partners” this is a very important point and I think it would be really good if you could add some reflections on what the literature is saying about this, even if there is not much. Then in the discussion you could see if your results and analysis are similar in anyway etc.

Response-11: We reviewed some literature on the issue (though we did not find many studies that give comprehensive explanation, see: page 5-6, lines 126-133; and also provided our speculation in the discussion in line with some of our findings, page 20- 21, lines 442-457.

Here, we have copied and pasted the whole paragraph included in the literature review section; page 5-6, as follows:

Studies have also consistently shown that childhood exposure to parental abuse is linked to their adulthood beliefs/behaviors related to wife abuse [23, 24], although little is known about why many adults who were exposed to childhood abusive events do not continue the same pattern with their partners [26]. It is possible that higher education and an awareness of the risks associated with abuse, a person’s self-esteem, relationship quality, meaningful victim support system, gender equitable environment, and good social control mechanism may de-motivate people to continue the abuse in adulthood even if they have experienced/internalized the abuse in their childhood [27-29]. A few studies based in Bangladesh have focused on the social learning of wife abuse as well as its social acceptance among community members [7, 30]. However, the association between childhood non-experience of various types of family victimization with attitudes in adulthood toward wife abuse is relatively overlooked in the previous studies conducted in Bangladesh. Hence, this study aims to enhance our understanding of what influence childhood learning and experiences of different abuses have on refusal of wife abuse in adulthood.

Comment-12:

Methods:

The methods section could be reduced quite a bit.

Response-12: We reduced the section accordingly.

Comment-13:

Study participants, you state clearly it was adults under 60, please indicate the youngest age for inclusion.

Response-13: We indicated it clearly; page 8, line 177. The same is reflected in the tables.

Comment-14:

 “We excluded men/women with mental/physical disabilities.” Please indicate why they were excluded.

Response-14: We have now mentioned why we excluded them; see page 8, lines 177-179.

We added this sentence: Due to our lack of preparation and appropriate facilities, we excluded individuals with mental disorders and physical disabilities (e.g., deaf/mute).

Comment-15:

‘Social Acceptance of Wife Abuse Scale’ (SAWAS) – its great that you used a locally developed and validated scale.

The childhood exposure questions – please specify whether these covered both experience of and witnessing abuse (I assume it did?), and if it includes witnessing as well, perhaps include an example of one.

Response-15: We only assessed the actual experience of abuse during their childhood, not the witness of inter-parental abuse. We have now made it clear in the paper; page 9, lines 217-220.

Comment-16:

Results

I think it is important to discuss the results by men and women’s experiences – I think it is important to understand how these views and experiences differ. It would also be interesting to discuss the analysis by age, which would show, among other things, if there are any generational shifts occurring etc. Please can you undertake and present these results as well. Its interesting to see the effect of education and ethnicity and I imagine gender and age would also reveal interesting findings.

Response-16: We have now presented the gender disaggregated findings on both their views and childhood experiences; see: page 13, lines 297-302, 304 (table 1), 306-313; page 14, 319 (table 2). We have also presented the gender and age specific findings on the ARWA; page 14, lines 333-336. Multivariate findings also addressed the issues of gender and ethnicity: page 17, lines 381-390. Younger people refused physical abuse more than the older people, page 18, line 391(table 4).

Comment-17:

Discussion

I feel that discussions should not try to present all the data again, and should make a relatively smooth, yet powerful, read, so I think you might want to re-write the first paragraph, could you shorten it to make it more readable. The final sentence in the first paragraph does a great job of summarizing some results very effectively, perhaps you could consider something like that for the beginning of the paragraph

Response-17: We have re-written the first paragraph. Now, we only address the key findings in the beginning of the discussion; page 19, lines 395-401. We revised the paragraph accordingly.

Comment-18:

 “These findings have offered a novel understanding of the issue and added to the current literatures that not all childhood abusive experiences equally contribute to the adulthood attitudinal refusal/acceptance of wife abuse” This is an important finding, and if we could also have this broken down by gender and age we would learn so much more!

Response-18: We have added a few lines on these findings as well as our understanding of this; page 20, lines 421-425. We observed that women’s attitudes were generally influenced by their childhood exposure to emotional abuse, while men’s attitudes were shaped by their experience of physical abuse; page 20, lines 423-425. We also discussed this according to formal educational attainment: page 20-21, lines 442-457. 

Comment-19:

Please consider a final edit of the paper as there are still a few grammatical errors.

Response-19: The paper has been edited by a professional editor.

Others: The findings from multivariate are now presented more systematically: five models for the ARWA are presented separately one by one: page 16-17, lines 347-380.

---

## [Decision Letter · Decision Letter 1]

3 Mar 2021

PONE-D-20-30007R1

Does childhood experience of family victimization influence adulthood refusal of wife abuse?

Evidence from rural Bangladesh

PLOS ONE

Dear Dr. Karim,

Thank you for submitting your manuscript to PLOS ONE. After careful consideration, we feel that it has merit but does not fully meet PLOS ONE’s publication criteria as it currently stands. Therefore, we invite you to submit a revised version of the manuscript that addresses the points raised during the review process.

We look forward to receiving your revised manuscript.

Kind regards,

Kristin Dunkle

Academic Editor

PLOS ONE

Journal Requirements:

Reviewers' comments:

Reviewer's Responses to Questions

**Comments to the Author**

1. If the authors have adequately addressed your comments raised in a previous round of review and you feel that this manuscript is now acceptable for publication, you may indicate that here to bypass the “Comments to the Author” section, enter your conflict of interest statement in the “Confidential to Editor” section, and submit your "Accept" recommendation.

Reviewer #1: (No Response)

Reviewer #2: (No Response)

2. Is the manuscript technically sound, and do the data support the conclusions?

Reviewer #1: Partly

Reviewer #2: Yes

3. Has the statistical analysis been performed appropriately and rigorously? 

Reviewer #1: No

Reviewer #2: Yes

4. Have the authors made all data underlying the findings in their manuscript fully available?

Reviewer #1: Yes

Reviewer #2: Yes

5. Is the manuscript presented in an intelligible fashion and written in standard English?

Reviewer #1: Yes

Reviewer #2: Yes

6. Review Comments to the Author

Reviewer #1: The authors have made some substantial revision and incorporated most of the recommendation from the previous comments and suggestions. I have a few more comments as follows:

1) Authors have indicated that the sample size formula indicated in the paper is for the main study which they have described as a cluste-randomised control trial. This makes their explanation even more confusing. The sample size calculation formula quoted in the paper does not seem to be for a cluster-randomised control trial that is supposed to assess effect of an intervention, but rather for a prevalence study. Could the authors clarify this.

2) Tables 1 and 2 has showed significant differences in both the outcomes and main exposures between men and women. Adjusted for sex of participant in the model may mask the relationship between childhood victimization and the outcome. It is also common practice in violence analysis to look at men and women separately. Authors need to discuss the differences in outcome and main exposure between men and women in the context of Bangladesh as well as in the context of existing knowledge.

3) Authors need to clarify further how the derived the 2 outcomes (refusal of emotional abuse and refusal of physical abuse) from the 6 items. It is clearer now how they derived the other 3 outcomes but it is not clear how participants’ responses to the 6 items were used to derive the 2 outcomes. Perhaps they could include a footnote as they have done for the exposure definition table.

4) Considering that the focus of the paper is on the relationship between childhood victimization and attitudes towards abuse of wife, the authors should present the final table of results that highlight this objective. I recommend that they present the unadjusted and then the adjusted ORs for the 2 exposures. They could then assess whether it is necessary to control for all the variable they have included in the adjusted model, considering that the number of participants in “severe abuse” categorical for some of the outcomes is small.

5) Table 3 should also be stratified by sex to complement Tables S2 and S3.

6) Have the authors looked at the intersection of childhood emotional abuse and physical abuse, considering that these are not mutually exclusive experiences.

7) Could the authors check why they are getting weird CIs for income in S3. A table of frequencies (as suggested in point 5) would assisting in understanding what is happening here.

Reviewer #2: Thank you for revising the paper in line with the peer review you received. I think the paper is greatly improved and presents a very important argument, the discussion is very well structured and makes many important points. However, I believe there are still a few small areas in which the paper could be improved, I hope my comments are helpful.

Introduction:

The first sentence reads as “Attitudinal (not actual act) refusal of wife abuse (ARWA) among men and women is a crucial factor that practitioners attempt to enhance in the society for the primary prevention of intimate-partner violence (IPV) in many low- and lower-middle income countries” I think you can delete the brackets (Not actual act) and make this point clearly in the third sentence, as the first sentence is written currently with the brackets is a little tricky to follow. Perhaps you could add to the end of your third sentence “… and focuses on attitudes and not actual acts of abuse.”

“In contrast, the refusal of wife abuse signifies the notions where both the male and the female members of a society reject the abuse [5]. This refers to the attitudinal stand where people reject wife abuse. ARWA is crucial in preventing IPV.” I think these sentences could be deleted, they are obvious and feel like you are over-stating the case and becoming repetitive.

” Violent husbands often try to defend the abuse by pointing out their ‘wives’ faults’” I’m not sure they are ‘defending their abuse’ as they probably don’t see it as abuse – I think you should rather talk about ‘explaining their actions’ – this does not imply that they think they need to defend anything nor that they think of it as abuse.

Theoretical perspective:

The first paragraph is repetitive, please remove the repetition. Perhaps it would be useful to go through the paragraph and see how many ideas/points you have in the paragraph and where you repeat the same idea or point, and then see what can be removed.

“however, a recent study indicates that there may be at least four dimensions where a person may refuse wife abuse on the grounds of: the wife failing to fulfill her family obligations, for challenging male family authority, physical abuse, and emotional abuse [31].” I think this sentence is wrong – I believe you are listing the reasons people use to justify men abusing their wives, I don’t think it is used as grounds to refuse abuse. I think you need to re-write this something like… “however, a recent study indicates that there may be at least four dimensions where a person may justify wife abuse on the grounds of: the wife failing to fulfill her family obligations, for challenging male family authority, physical abuse, and emotional abuse.”

Control variables:

You note that “They [Bengali’s] are traditionally patriarchal. In contrast, the Garos are an ethnic minority community upholding matrilineal traditions, whereas the Santals are a patriarchal indigenous community living in Bangladesh”. I believe this is a very important point about the communities you are studying, and it is important to understand whether differences occur for the matrilineal communities, therefore I think you should also mention this when describing the study sites earlier.

Results:

I would suggest you consider moving the socio-demographic details of the participants to the beginning of the results, It is often good to begin by describing your participants (with the socio-demographic detail) and then move into other results.

“Our findings also reveal that respondents from ethnic communities are more likely to refuse most of the wife abuses than the mainstream communality”, please clarify who you are referring to when you say ‘ethnic communities’, I assume you mean the two communities who are no Bengali? Please clarify.

Discussion:

“Specifically, our study indicates that the childhood exposure to mild emotional abuse might increase the attitudinal refusal of wife abuse in adulthood” this sentence is a bit confusing – am I right to think that exposure to ‘mild emotional abuse’ leads to adults being more likely to accept wife abuse, ie not to refuse wife abuse? Please clarify the sentence.

Grammar:

I note you had the paper reviewed for grammar, and I think it is reading much better, but I think it could benedict from one more edit, there are places where the sentence structure is difficult to follow and this can confuse the reader, for example in the results you say “The attitudinal refusal of different wife abuses also appeared to be less among women than men” I believe it might read better if written as “The attitudinal refusal of different FORMS OF wife abuse appeared to be LOWER among women than men” Another example is that in the results the authors shift to using the acronym ARWA a lot, I would suggest rather writing ‘attitudinal refusal of wife abuse’ I think using the acronym interrupts the flow of the sentences, however, I see that this may become very repetitive, so perhaps the sentences could be restructured to address repetition if needed.

7. PLOS authors have the option to publish the peer review history of their article (what does this mean?). If published, this will include your full peer review and any attached files.

Reviewer #1: No

Reviewer #2: No

---

## [Author Response · Author response to Decision Letter 1]

22 Mar 2021

Thank you very much both the reviewers and the editor.

We have addressed all of your comments as follows:

The changes are track-changed/highlighted as colored text throughout the paper. 

Comments

Journal Requirements:

Response:

We have removed the following reference and replaced it with an original paper: 

 “25. Gelles RJ. Spouse Abuse - Theoretical Explanations. Marriage and Family Encyclopedia: Net Industries; 1995 [October 15, 2007]. Available from: http://family.jrank.org/pages/1629/Spouse-Abuse-THEORETICAL-EXPLANATIONS.html.

Replaced with: “25. Gelles RJ. Family Violence. Annu Rev Sociol. 1985; 11: 329-346.”

Comments

Reviewer #1: 

The authors have made some substantial revision and incorporated most of the recommendation from the previous comments and suggestions. I have a few more comments as follows:

1) Authors have indicated that the sample size formula indicated in the paper is for the main study which they have described as a cluster-randomised control trial. This makes their explanation even more confusing. The sample size calculation formula quoted in the paper does not seem to be for a cluster-randomised control trial that is supposed to assess effect of an intervention, but rather for a prevalence study. Could the authors clarify this.

Response 1: 

We have further revised the sentences and made it clear (see page 8, line: 213-217) as: 

“By taking into account the findings from our pilot study conducted in selected villages, we computed the sample size for the baseline survey. Considering the prevalence of sexual abuse in a village (p=0.167), the minimum sample size was estimated at 1,854 using a formula (n = z2α/2 p(1 p)/E2…” 

Our further replies:

Regarding the effects of community mobilization intervention on domestic violence, we lack previous data in the context of Bangladesh. Therefore, in order to estimate the required minimum sample size, we used the current prevalence of DV in a study village. We tried to maximize the required sample size and that is why we used the lowest prevalence of sexual abuse.

Now we have data on our intervention effects in a few Bengali villages. It shows that the prevalence of emotional DV in our intervention sites is 72% while it is 81% in the control areas. Thus if we estimate our minimum sample size for each group, it should be 472 for 95% confidence interval and 90% given power (e.g. using the formula: sample size: n= (Z1-β+Z1-α/2)2[(R+1)]�p2(R2+1)/p2(1-R)2 ; where n=sample size in each group, p1=proportion in intervention group, p2=proportion in control group, R= relative risk, Z1-β = Z score for 90 percentile, and Z1-α/2 = Type 1 error set at α=0.05. Adjusting with 95% response rate in the intervention, thus it should be enough to have total 500+500= 1000 samples from the villages. Thus, we believe that our 1,929 samples should be representative for both prevalence and intervention studies.

Comments

2) Tables 1 and 2 has showed significant differences in both the outcomes and main exposures between men and women. Adjusted for sex of participant in the model may mask the relationship between childhood victimization and the outcome. It is also common practice in violence analysis to look at men and women separately. Authors need to discuss the differences in outcome and main exposure between men and women in the context of Bangladesh as well as in the context of existing knowledge.

Response 2: 

We are keeping the current model as it helps us to understand the gender differences in the attitudinal refusal of wife abuse, though we are aware of the limitations of this strategy. However, we have also presented the findings for men and women separately in S2-3 Tables.

Considering the context of Bangladesh as well as existing knowledge, we also discussed the issue why only few women attitudinally refused wife abuse compared to men as follows: 

“The study further shows that men are more likely to refuse all types of wife abuse compared to women in rural Bangladesh. These results are in line with the findings of previous studies showing that women uphold more conservative gender ideology than their male counterparts in low-income patriarchal societies like Bangladesh [11, 12, 45]. This is possible because both men and women can contribute to accept, maintain and reinforce the patriarchal social order. As a ‘subordinated gender’, women’s acceptance/reinforcement of these patriarchal traditions makes sure the continuation of wife abuse/male domination over women [12, 54]. This indicates that most of the women are not so aware of their rights.” 

 (See page 25, Line: 496-503)

Comments

3) Authors need to clarify further how the derived the 2 outcomes (refusal of emotional abuse and refusal of physical abuse) from the 6 items. It is clearer now how they derived the other 3 outcomes but it is not clear how participants’ responses to the 6 items were used to derive the 2 outcomes. Perhaps they could include a footnote as they have done for the exposure definition table.

Response 3: 

We have further clarified this in the method section as follows (see page 9, line: 237-241): 

 “In order to compute the variables (refusal of emotional abuse and refusal of physical abuse), we recoded the responses. We derived the refusal of emotional abuse variable after merging ‘yes – emotional abuse’ and ‘yes – both emotional and physical abuse’ responses. Then again, we got the refusal of physical abuse by adding ‘yes – physical abuse’ and ‘yes – both emotional and physical abuses’ responses.”

We have also added a footnote under Table 1 (see page 14, line: 348-352).

Comments

4) Considering that the focus of the paper is on the relationship between childhood victimization and attitudes towards abuse of wife, the authors should present the final table of results that highlight this objective. I recommend that they present the unadjusted and then the adjusted ORs for the 2 exposures. They could then assess whether it is necessary to control for all the variable they have included in the adjusted model, considering that the number of participants in “severe abuse” categorical for some of the outcomes is small.

Response 4: 

We have now presented unadjusted ORs in the Table 3 (see page 22).

In the adjusted models, we have only included the significant variables (see Table 4) (page 23).

Comments

5) Table 3 should also be stratified by sex to complement Tables S2 and S3.

Response 5: 

Part of Table 3, Table 1 and Table 2 are now merged and presented as Table 1 (page 14). 

The Table 1 is now presenting sample socio-demographic characteristics, and the prevalence of outcome variables as well as the prevalence of exposure variables among the samples. 

Table 3 is now presented as Table 2 (page 17-18). 

All of these tables (Table 1 and Table 2) are now stratified by sex. 

Comments

6) Have the authors looked at the intersection of childhood emotional abuse and physical abuse, considering that these are not mutually exclusive experiences.

Response 6: 

We have found no significant interactions between these two variables. Thank you so much. 

Comments

7) Could the authors check why they are getting weird CIs for income in S3. A table of frequencies (as suggested in point 5) would assisting in understanding what is happening here.

Response 7:

We have corrected this. It was related to a category of income, where we had only two samples. They could be the outliers in the male samples; e.g., two men did not have any income during our fieldwork. So, we have merged ‘No income’ and ‘Bellow BDT 7000” categories (see S3 Table). 

Comments

Reviewer #2: 

Thank you for revising the paper in line with the peer review you received. I think the paper is greatly improved and presents a very important argument, the discussion is very well structured and makes many important points. However, I believe there are still a few small areas in which the paper could be improved, I hope my comments are helpful.

Introduction:

The first sentence reads as “Attitudinal (not actual act) refusal of wife abuse (ARWA) among men and women is a crucial factor that practitioners attempt to enhance in the society for the primary prevention of intimate-partner violence (IPV) in many low- and lower-middle income countries” I think you can delete the brackets (Not actual act) and make this point clearly in the third sentence, as the first sentence is written currently with the brackets is a little tricky to follow. Perhaps you could add to the end of your third sentence “… and focuses on attitudes and not actual acts of abuse.”

Response 8: 

We made these changes (see page 3, line 59-62).

The third sentence now read as “As the attitudinal refusal of wife abuse focuses on a mindset and not actual acts of abuse, it is a crucial aspect of IPV. Refusal/acceptance of wife abuse is also a widely talked about issue in the discourse of the primary prevention of IPV against women.”

Comments

 “In contrast, the refusal of wife abuse signifies the notions where both the male and the female members of a society reject the abuse [5]. This refers to the attitudinal stand where people reject wife abuse. ARWA is crucial in preventing IPV.” I think these sentences could be deleted, they are obvious and feel like you are over-stating the case and becoming repetitive.

Response 9:

We deleted these sentences as the theme appears in the previous sentences (page 3, line 69). 

Comments

” Violent husbands often try to defend the abuse by pointing out their ‘wives’ faults’” I’m not sure they are ‘defending their abuse’ as they probably don’t see it as abuse – I think you should rather talk about ‘explaining their actions’ – this does not imply that they think they need to defend anything nor that they think of it as abuse.

Response 10:

We revised the sentence (see page 4, line 107) as: 

“Violent husbands often try to explain their actions by pointing out their ‘wives’ faults’”

Comments

Theoretical perspective:

The first paragraph is repetitive, please remove the repetition. Perhaps it would be useful to go through the paragraph and see how many ideas/points you have in the paragraph and where you repeat the same idea or point, and then see what can be removed.

Response 11:

We have now removed all the repetitions from the paragraph (see page 5: line 120-134).

Comments

 “however, a recent study indicates that there may be at least four dimensions where a person may refuse wife abuse on the grounds of: the wife failing to fulfill her family obligations, for challenging male family authority, physical abuse, and emotional abuse [31].” I think this sentence is wrong – I believe you are listing the reasons people use to justify men abusing their wives, I don’t think it is used as grounds to refuse abuse. I think you need to re-write this something like… “however, a recent study indicates that there may be at least four dimensions where a person may justify wife abuse on the grounds of: the wife failing to fulfill her family obligations, for challenging male family authority, physical abuse, and emotional abuse.”

Response 12:

We have revised the sentences (see page 6, line 163-167) as: 

“however, a recent study indicates that there may be at least four dimensions where a person may justify/refuse wife abuse [31]. The four dimensions are as follows: acceptance/refusal of emotional abuse, acceptance/refusal of physical abuse, acceptance/refusal of abuse on the grounds of wife’s disobeying family obligations, and acceptance/refusal of abuse on wife’s challenging male family authority, [31].”

Comments

Control variables:

You note that “They [Bengali’s] are traditionally patriarchal. In contrast, the Garos are an ethnic minority community upholding matrilineal traditions, whereas the Santals are a patriarchal indigenous community living in Bangladesh”. I believe this is a very important point about the communities you are studying, and it is important to understand whether differences occur for the matrilineal communities, therefore I think you should also mention this when describing the study sites earlier.

Response 13:

We added this under the section ‘Sites under study’ (see page 7, line: 200-207): 

The text can be read now as follows:

“The mainstream Bengali and the ethnic minority Santal villages are patriarchal whereas the ethnic Garo society follows matrilineal (matriarchal) traditions. In the Garo villages, women are traditionally treated as the heads of the households; and all of the family properties are also passed down to the female line. On the other hand, men own most of the household properties as well as they rule the families in the both Bengali and Santal villages. Due to their veiled seclusion, women’s physical mobility is quite low in the Bengali villages. However, women enjoy greater freedom to move in public spheres in the both Garo and Santal villages.”

Comments

Results:

I would suggest you consider moving the socio-demographic details of the participants to the beginning of the results, It is often good to begin by describing your participants (with the socio-demographic detail) and then move into other results.

Response 14:

We have now started by presenting the socio-demographic details (page 13, line 336). 

Tables are also now rearranged accordingly. Please see: page 14, 17, 22 and 23)

Comments

 “Our findings also reveal that respondents from ethnic communities are more likely to refuse most of the wife abuses than the mainstream communality”, please clarify who you are referring to when you say ‘ethnic communities’, I assume you mean the two communities who are no Bengali? Please clarify.

Response 15:

We have revised the sentence (page 26, line 504-506) as:

“Our findings also reveal that respondents from the Garo and Santal ethnic minority communities are more likely to refuse most types of wife abuse than the mainstream Bengali communality.”

Comments

Discussion:

“Specifically, our study indicates that the childhood exposure to mild emotional abuse might increase the attitudinal refusal of wife abuse in adulthood” this sentence is a bit confusing – am I right to think that exposure to ‘mild emotional abuse’ leads to adults being more likely to accept wife abuse, ie not to refuse wife abuse? Please clarify the sentence.

Response 16:

We have clarified the sentence (see page 25, line 485-487) as:

“Regarding the childhood victimization of emotional abuse, our study indicates that those who experienced mild emotional abuse during childhood are more likely to refuse the wife abuse in adulthood than those who experienced severe emotional abuse.”

We also surprised with this finding, however we provided an explanation (page 25, line 347-495)

Comments

Grammar:

I note you had the paper reviewed for grammar, and I think it is reading much better, but I think it could benedict from one more edit, there are places where the sentence structure is difficult to follow and this can confuse the reader, for example in the results you say “The attitudinal refusal of different wife abuses also appeared to be less among women than men” I believe it might read better if written as “The attitudinal refusal of different FORMS OF wife abuse appeared to be LOWER among women than men” Another example is that in the results the authors shift to using the acronym ARWA a lot, I would suggest rather writing ‘attitudinal refusal of wife abuse’ I think using the acronym interrupts the flow of the sentences, however, I see that this may become very repetitive, so perhaps the sentences could be restructured to address repetition if needed.

Response 17:

We have gone through the paper again for further language edit and we tried to address the issues you raised. We also replaced the acronym ARWA with its full form. Thank you.

---

## [Decision Letter · Decision Letter 2]

3 May 2021

PONE-D-20-30007R2

Does childhood experience of family victimization influence adulthood refusal of wife abuse?

Evidence from rural Bangladesh

PLOS ONE

Dear Dr. Karim,

Thank you for submitting your manuscript to PLOS ONE. After careful consideration, we feel that it has merit but does not fully meet PLOS ONE’s publication criteria as it currently stands. Therefore, we invite you to submit a revised version of the manuscript that addresses the points raised during the review process.

We look forward to receiving your revised manuscript.

Kind regards,

Kristin Dunkle

Academic Editor

PLOS ONE

Journal Requirements:

Reviewers' comments:

Reviewer's Responses to Questions

**Comments to the Author**

1. If the authors have adequately addressed your comments raised in a previous round of review and you feel that this manuscript is now acceptable for publication, you may indicate that here to bypass the “Comments to the Author” section, enter your conflict of interest statement in the “Confidential to Editor” section, and submit your "Accept" recommendation.

Reviewer #1: (No Response)

Reviewer #2: All comments have been addressed

2. Is the manuscript technically sound, and do the data support the conclusions?

Reviewer #1: Yes

Reviewer #2: Yes

3. Has the statistical analysis been performed appropriately and rigorously? 

Reviewer #1: Yes

Reviewer #2: Yes

4. Have the authors made all data underlying the findings in their manuscript fully available?

Reviewer #1: Yes

Reviewer #2: Yes

5. Is the manuscript presented in an intelligible fashion and written in standard English?

Reviewer #1: Yes

Reviewer #2: Yes

6. Review Comments to the Author

Reviewer #1: The authors have made some substantial revision and incorporated most of the recommendation from the previous comments and suggestions. However, I still have a few more comments, suggestions and request for clarification as follows:

i)With the sample consisting of both men and women and also from 2 different traditions (matrilineal and patrilineal), I wonder how the 2 (sex of participant and type of tradition) are interacting. Did the authors look at the interaction between sex of participant and type of tradition or ethnic grp?

ii)Could the authors clarify in the paper how a participant was selected in a household where both a man and woman were available.

iii)I would like the authors to clarify more explicitly either in the text or in Table S1 how they are classifying the 4 responses for each item into refusal and non-refusal for the two outcomes ( refusal of emotional abuse and refusal of physical abuse). I still find it hard to understand how these 2 outcomes were derived. For example, when they are looking at refusal of emotional abuse, does the ‘refusal group’ include the ‘yes, physical abuse’ ( does it include those that responded 0=no abuse to all the 6 items plus those that responded 2=yes, physical abuse to all the 6 items?

iv)In line 287-289, the authors have defined the OR with respect to a continuous exposure when none of the exposures used in their models are continuous. I would recommend that they just state that ORs were used to report associations between exposures and outcomes.

v)In line 190, instead of “To avoid for possible non-consent and drop out…” they should write “To account for …..”

vi)In line 348, change “not-victims’ to ‘non-victims’

vii)Could the authors discuss the any differences in associations between outcomes and ethnicity between men and women ( Table S2 & S3).

Reviewer #2: (No Response)

7. PLOS authors have the option to publish the peer review history of their article (what does this mean?). If published, this will include your full peer review and any attached files.

Reviewer #1: No

Reviewer #2: No

---

## [Author Response · Author response to Decision Letter 2]

5 May 2021

Dear Dr. Kristin Dunkle,

Thank you very much.

We have addressed all of the comments as follows:

The changes are track-changed/highlighted as colored text throughout the paper. 

We have tested the interactions between the sex of study participants and the type of ethnic tradition (patrilineal or matrilineal) and found that none of them were statistically significant. We reported the test of model effects in detail in a separate file, “Response to Reviewers.”

We have further clarified the sampling procedure - how we have selected a participant from a household (line 191-199). Please also see in the “Response to Reviewers.”

We have clarified how we derived the two outcomes, refusal of emotional abuse and refusal of physical abuse, in Table S1. The table represents how we have recoded the data and classified the variables. Please see Table S1.

As the reviewer suggested, we have revised two sentences: Line: 292-293, Line: 188-189

We have also changed the word from “not-victims’ to ‘non-victims.’ Line: 352

We have discussed the gender differences in associations between outcomes and ethnicity in the three sections of the manuscript: Result section: line 420-432; Discussion section: line 498-594; and Limitation section: line 535-539

These are all the changes we made in the manuscript and Table S1. 

Details of the changes made are reported in the file “Response to Reviewers.”

Best regards,

Rabiul

---

## [Editor Report · Decision Letter 3]

19 May 2021

Does childhood experience of family victimization influence adulthood refusal of wife abuse?

Evidence from rural Bangladesh

PONE-D-20-30007R3

Dear Dr. Karim,

We’re pleased to inform you that your manuscript has been judged scientifically suitable for publication and will be formally accepted for publication once it meets all outstanding technical requirements.

Kind regards,

Kristin Dunkle

Academic Editor

PLOS ONE
---

## [Editor Report · Acceptance letter]

25 May 2021

PONE-D-20-30007R3 

Does childhood experience of family victimization influence adulthood refusal of wife abuse?Evidence from rural Bangladesh 

Dear Dr. Karim:

I'm pleased to inform you that your manuscript has been deemed suitable for publication in PLOS ONE. Congratulations! Your manuscript is now with our production department. 

Kind regards, 

on behalf of

Dr. Kristin Dunkle 

Academic Editor

PLOS ONE